



# Difficulties in explaining complex issues with maps.
# Evaluating seismic hazard communication – the Swiss case

Michèle Marti[1], Michael Stauffacher[2], Stefan Wiemer[1]

[1]Swiss Seismological Service, ETH Zurich, Zurich, 8092, Switzerland
[2]USYS TdLab, ETH Zurich, Zurich, 8092, Switzerland

*Correspondence to*: Michèle Marti (michele.marti@sed.ethz.ch)

**Abstract**

2.7 billion people live in areas where earthquakes causing at least slight damage have to be expected regularly. Providing
information can potentially save lives and improve the resilience of a society. Maps are an established way to illustrate natural hazard. Despite of being mainly tailored to the requirements of professional users, they are often the only accessible information to help the public deciding about mitigation measures. There is evidence that hazard maps are frequently misconceived. Visual and textual characteristics as well as the manner of presentation have been shown to influence their comprehensibility. Using a real case, the material to communicate the updated seismic hazard model for Switzerland was
analyzed in a representative online survey of the population (N = 491) and in two workshops involving architects and engineers not specializing in seismic retrofitting (N = 23). Although many best practice recommendations have been followed, the understanding of seismic hazard information remains challenging. Whereas most participants were able to distinguish hazardous from less hazardous areas, correctly interpreting detailed results and identifying the most suitable set of information for answering a given question proved demanding. We suggest scrutinizing current natural hazard communication strategies
and empirically testing new products.

## 1 Introduction

The preferred means of communicating complex natural hazard calculations are currently maps (Bostrom et al., 2008; Gaspar-Escribano and Iturrioz, 2011; Kunz and Hurni, 2011). Even though natural hazard maps are mainly tailored to the needs of primary users (Perry et al., 2016), they are used unaltered to communicate with other recipients (Thompson et al., 2015). In
consequence, recent publications indicate that they often fail to transmit their content (Meyer et al., 2012). Non-experts in the field, in particular, often struggle to interpret the maps correctly (Hagemeier-Klose and Wagner, 2009; Kjellgren, 2013; Perry et al., 2016; Severtson and Vatovec, 2012). This is fundamental, as improving resilience requires not only knowledgeable experts, but also politicians, authorities, and an informed public to support precautionary actions.

2.7 billion people live in areas where earthquakes causing at least slight damage have to be expected regularly[1] (Pesaresi et al.,

2017). Earthquakes cannot be predicted, therefore knowing and understanding seismic hazard is a major step towards loss reduction (Gaspar-Escribano and Iturrioz, 2011; Shaw et al., 2004). It enables a society to take precautionary measures like persisting in the application of building codes or securing movable items. An earthquake-resistant building design, based on seismic hazard values (Perry et al., 2016), is the most efficient means of reducing seismic risk.

Identifying and providing seismic hazard values is a primary responsibility of seismological services around the world.

Earthquake hazard describes how often a certain horizontal acceleration caused by an earthquake has to be anticipated at a specific location (Swiss Seismological Service, 2018). The most prominent output of such seismic hazard assessments are maps, which are often the only accessible information to help the public deciding about mitigation measures (see a selection in Fig. 1).

| a) Switzerland, 2014 | b) USA |
| --- | --- |

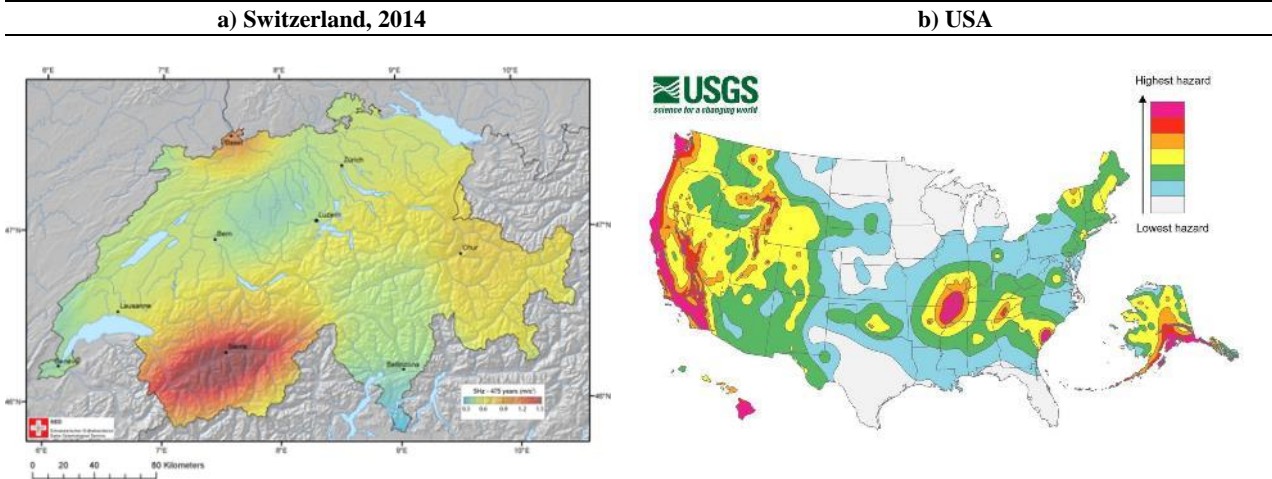





[1] The global seismic hazard map (EMMI-GSHAP) defines areas as hazardous if there is a 10 % chance of exceedance in 50 years for earthquakes with a minimal intensity of V on the Mercalli scale.

| c) France | d) Canada |
|---|---|

| e) Italy | f) New Zealand |
|---|---|

**g) Global Earthquake Model**

**Fig. 1.** Selection of national seismic hazard maps: a) Swiss seismic hazard map (Swiss Seismological Service, 2018, www.seismo.ethz.ch/knowledge/seismic-hazard-switzerland/) b) US seismic hazard map (United States Geological Survey, 2018, https://earthquake.usgs.gov/hazards/hazmaps/) c) Zonage sismique de la France (Bureau de recherches géologiques et minières, 2018,



http://www.planseisme.fr/Zonage-sismique-de-la-France.html) d) Simplified seismic hazard map for Canada, the provinces and territories (Natural Resources Canada, 2018) e) Pericolosità sismica di riferimento per il territorio nazionale (Istituto Nazionale di Geofisica e

Vulcanologia, 2018, http://zonesismiche.mi.ingv.it/) f) The 2010 National Seismic Hazard Model for New Zealand (Institute of Geological and Nuclear Sciences Limited, 2018), https://www.gns.cri.nz/Home/Our-Science/Natural-Hazards/Earthquakes/Earthquake-Forecast-and-Hazard-Modelling/2010-National-Seismic-Hazard-Model g) Global Earthquake Model (GEM) Seismic Hazard Map (version 2018.1 - December 2018) (Pagani et al., 2018, https://www.globalquakemodel.org/gem).

The main users or recipients of seismic hazard maps can be broken down into three groups (Meyer et al., 2012): (1) experts,

mainly seismologists, geologists, and specialized civil engineers, who use seismic hazard maps on a regular basis for professional purposes; (2) other professionals, like architects, engineers not specializing in seismic retrofitting, and emergency and disaster managers, who only deal occasionally with seismic hazard maps; (3) the public, who are confronted by authorities or media with seismic hazard maps or seek for advice before purchasing a house or contracting an insurance. They are usually unfamiliar with many of the maps components'.

Previous studies evaluating maps for risk management purposes mainly focused on directly-involved stakeholders and authorities (Dransch et al., 2010). The few studies that analyzed the public's needs regarding hazard maps did so either by questioning experts or by mostly relying on a small sample (Hagemeier-Klose and Wagner, 2009; Kjellgren, 2013; Meyer et al., 2012; Thompson et al., 2015). The understanding of seismic hazard by non-experts in the field and the public has thus been neglected. The challenge lies not only in making accurate information available, but in presenting it in understandable ways

(Peters et al., 2008). Disseminating hazard maps online is seen as an important option of providing hazard information to the public (Hagemeier-Klose and Wagner, 2009; Kostelnick et al., 2013)

With respect to flood maps, Meyer et al. (Meyer et al., 2012) recommend implementing a less complex map design for the public in contrast to primary users. Different requirements and expectations are also emphasized by Hagemeier-Klose and Wagner (Hagemeier-Klose and Wagner, 2009), who stress that when presenting flood maps, technical terms should be avoided

and "emotional empathy" created. With regard to volcanic hazard maps, accurate data classification, meaningful application of color schemes, and textual elements are emphasized as being important for user engagement and the interpretation of map content (Thompson et al., 2015). Overall, there is a serious lack of empirically tested knowledge on how to design (seismic) hazard maps, especially when addressing the public.

This study takes a real-world setting to understand how seismic hazard maps provided by the Swiss Seismological Service

(SED) at ETH Zurich are read and understood by the general public and architects and engineers not specializing in seismic retrofitting. We focus on those recipients that are indispensable to improve earthquake resilience but are currently neither in the focus of the producers of most seismic hazard outputs nor of the research about maps to communicate natural hazards. There is also a genuine interest of this user groups, as the website access statistics of the SED show. The respective pages are among the most popular of the SED website. In addition, media often requests to reprint the seismic hazard map for a return

period of 475 years (see Fig. 2).



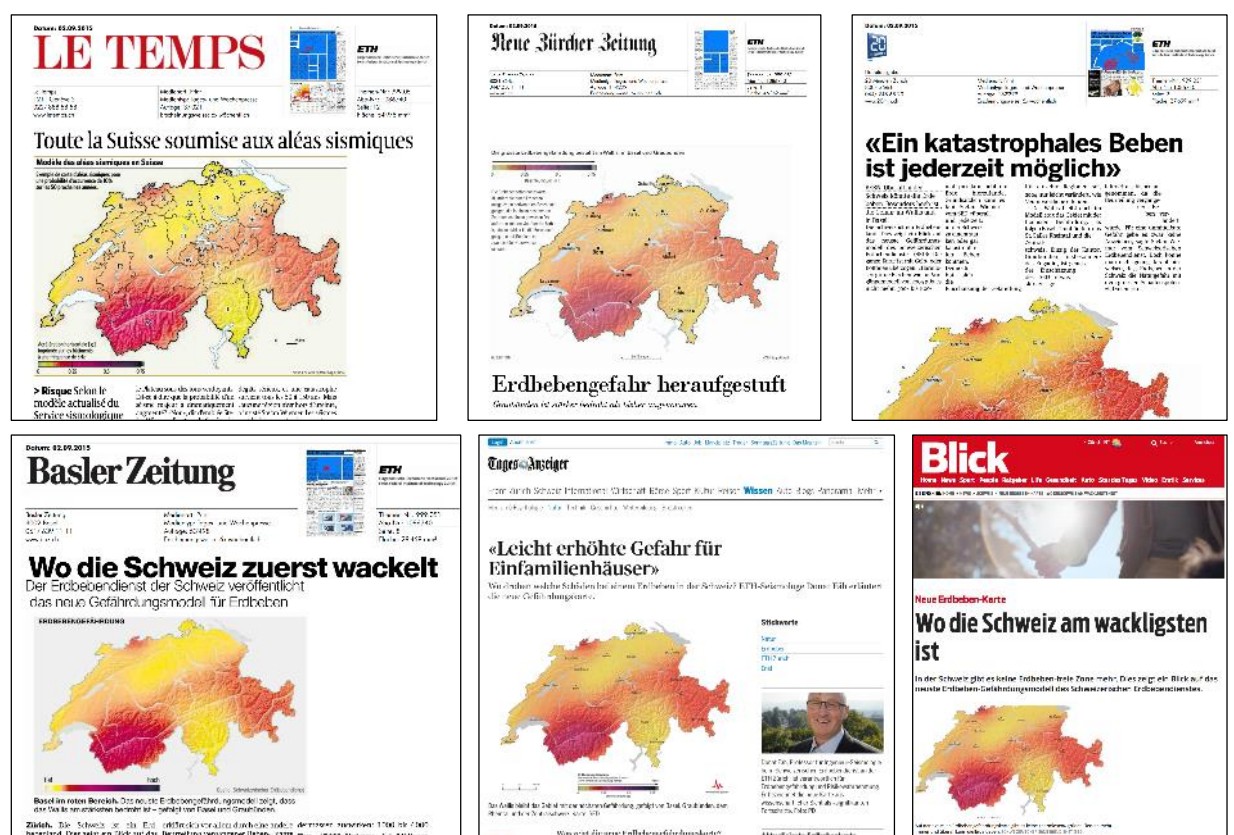

**Fig. 2.** Examples of reprints of the Swiss seismic hazard map for a return period of 475 years in popular national media outlets. Screenshots of Swiss newspapers taken by the authors.

This case was chosen as it is representative of the way results of a natural hazard assessment are presented to a variety of users. It typically reflects the work seismological services and other natural hazard agencies are doing around the world. By not only

taking into account the newest findings in the model calculation but also in the presentation of the results, the SED has gone one step further. We analyze how well participants are able to handle the information provided as well as their competence in deriving answers to given questions. We also examine their ability to interpret statistical information and the effect of interactive access. To our knowledge, for the first time, seismic hazard information is comprehensively tested in a real-world setting. The results will allow to derive best practices for improving seismic hazard and natural hazard communication

worldwide.

## 2 Best practices in communicating seismic hazard

Risk communication can lead to more accurate beliefs about seismic hazard and a higher tendency towards taking precautionary measures (Whitney et al., 2004). As elaborated previously, maps are the means of choice to communicate seismic





hazard. In the following, we discuss the factors determining how hazard maps are read, interpreted, and understood. This sets the baseline to analyze the maps produced by the SED.

## 2.1 Visual characteristics

Visual characteristics of seismic hazard maps are mainly defined by colors, contrast, and the explanatory legend. A survey compared volcanic hazard maps with a red-yellow and a red-yellow-blue color scheme (Thompson et al., 2015). Despite indicating the same values, identical hazard levels were interpreted differently. In a red-yellow map, areas colored yellow were considered to be at risk. Unlike, in a red-yellow-blue map, areas previously colored yellow and now colored blue gave the impression of being safe. Red color schemes are, with some cultural differences, commonly associated with danger, hazard, and risk (Bostrom et al., 2008). In contrast, light colors naturally seem less alarming than dark colors (Gaspar-Escribano and Iturrioz, 2011; Peters et al., 2008).

Clear colors and high contrast ratios improve the understanding of maps (Hagemeier-Klose and Wagner, 2009). Contrasts are especially relevant for people with defective color vision (Kunz et al., 2011). This is particularly important for maps with color schemes ranging from green to red (Thompson et al., 2015). Depicting certain values directly on the map instead of only mentioning them in the legend helps people with visual impairments to interpret the content correctly.

The chosen colors should allow easy distinction between data classes. The use of different color hues at each end of the scheme instead of a single hue helps. However, having too many classes diminishes users' ability to distinguish color values and decreases saturation of a specific class (Kunz and Hurni, 2011). It is worth testing which intervals are most likely to be understood and categorizing the data into three to five classes (Fuchs et al., 2011; Gaspar-Escribano and Iturrioz, 2011). Alternatively, unclassed maps can be used to depict continuous data. Even though users cannot distinguish small changes and might have difficulties in situating single data points in the legend, unclassed maps represent the data more accurately (Severtson and Myers, 2013).

Legends are another important aspect of visuals. If users cannot clearly understand or see the legend, they will probably misunderstand the map content (Kunz and Hurni, 2011). As different users have different needs, Gaspar-Escribano and Iturrioz (Gaspar-Escribano and Iturrioz, 2011) recommend comprehensive, numerical information for professionals and qualitative legends for non-professionals. Another approach suggests combining unclassed maps with verbal legends (e.g. low to high risk), as in any case, users struggle to assign specific color hues to single data points in the legend. In contrast, isarithmic maps, which connect points of equal values with lines, are preferably combined with numerical values (Severtson and Myers, 2013).

## 2.2 Textual characteristics

Descriptions support the understanding of graphics and enhance their persuasive impact (Lipkus, 2007). To reach this aim, the tradeoff between the completeness and the comprehensibility of information needs to be well balanced. Access to more complete information does not necessarily lead into enhanced comprehension and a better quality of choice (Peters et al., 2007). This is especially true for older persons and those with lower numeracy skills (Peters, 2008). Numeracy acts as



representative for cognition (Severtson and Myers, 2013) and may influence the general ability to understand graphics (Spiegelhalter et al., 2011). Providers should shift from an approach centered on information completeness to one that facilitates users' decision-making(Peters, 2008) by emphasizing important information (Pang, 2008).

In the context of seismic hazard communication, technical jargon, transmitting odds and other statistical information is further
of special relevance.

### 2.2.1 Technical vocabulary

Whenever possible, technical vocabulary should be avoided (Hagemeier-Klose and Wagner, 2009). Recent usability studies (e.g. (Burningham et al., 2008)) in the context of flood hazards emphasize that non-experts struggle to understand technical terms accompanying flood maps, like "return periods expressed as probabilities" (Meyer et al., 2012) or "one hundred year
flood" (Hagemeier-Klose and Wagner, 2008).

### 2.2.2 Odds

Most people struggle to understand odds. What they would like to know is the likelihood of an earthquake occurring within a conceivable period (Nathe, 2000). In the context of volcanic hazard, using "within" instead of "in" to describe the period helps to achieve a more balanced judgment of the distribution of likelihood of volcanic eruptions over a given time frame (Hudson-
Doyle et al., 2011). Nevertheless, the effect is only visible for longer periods and is more pronounced in likelihood judgments by non-scientists (Doyle et al., 2014).

### 2.2.3 Statistical information

When it comes to statistics, it has proven especially exigent to communicate single-event probabilities, conditional probabilities, and relative risks (Gigerenzer and Edwards, 2003).
To avoid misinterpretations of single-event probabilities and conditional probabilities, Statistical judgments by experts and non-experts improve similarly if they are based on frequencies rather than probabilities (Hoffrage et al., 2000). Nevertheless, in a study on volcanic hazard, participants with relatively high numeracy skills expressed a preference for percentages only, or percentages in combination with natural frequencies (Thompson et al., 2015). Alternatively, verbal and linguistic probabilities can be used (e.g. "likely", "certain"), even though they appear to be interpreted very differently. To minimize the
risk of misinterpretation, combining verbal and numerical information is seen as the most promising approach (Bodemer and Gaissmaier, 2012; Budescu et al., 2014).

Conditional probabilities pose another challenge. The standard seismic hazard map depicts a probability of exceedance of 10 percent within 50 years. Health-related studies demonstrate that such conditional probabilities are often misconceived by both physicians and patients (Bodemer and Gaissmaier, 2012; Gigerenzer and Edwards, 2003).





Relative risks are more difficult to understand than absolute risks (Bodemer and Gaissmaier, 2012; Gigerenzer and Edwards, 2003). Communicating absolute risks improves the correct understanding of a given statistical statement (Gigerenzer and Edwards, 2003).

## 2.3 Manner of presentation

Experiential and interactive information generates a stronger impact on attitudes and leads to a higher level of preparedness
(Becker et al., 2013; McIvor and Paton, 2007). Bostrom et al. (Bostrom et al., 2008) point out the potential in offering interactive visualizations to explore seismic risk information, allowing individual configurations to cover different user groups' needs. However, interactive visualizations should not be overloaded or too complex. Moreover, they should be based on clear communication goals and only offer functionalities that serve those (Dransch et al., 2010). In a study, natural hazard experts confirmed the usefulness of interactive hazard mapping tools (Kunz and Hurni, 2011).

Interactive map visualization facilitates the comparison of different parameters and allows personalized settings e.g. for transparency. Maps should enable appropriate hazard assessment and therefore make it possible to compare hazards at different times and in different areas (Dransch et al., 2010; Hagemeier-Klose and Wagner, 2009). Adaptive zooming is strongly recommended. It reduces the amount of information visible at once (Kunz and Hurni, 2011). The interface provided has to be user-friendly and offer access to further information (Hagemeier-Klose and Wagner, 2008). An extensive review of flood maps
in Europe revealed that all analyzed maps were either too simple or too complex. Many included too many functionalities and too much information, which diminished their comprehensibility (Hagemeier-Klose and Wagner, 2009).

## 3. Case study and focus of research

Testing hazard products is seen as an important success factor for information-presenting strategies (Kostelnick et al., 2013; Perry et al., 2016; Peters, 2008; Thompson et al., 2015). A testing campaign should determine how well the given information
is understood, the extent to which the communication goals are reached, and the influence of the presented materials on actual choices (Peters et al., 2008). Our cases study uses the original maps the SED provides to communicate its hazard model. In the following, we discuss their qualities with respect to the aforementioned best practices.

### 3.1 Qualities in the presentation of the Swiss seismic hazard model

Besides traditional hazard maps depicting ground acceleration values, the SED introduced two other map types: effect and
magnitude maps (see Fig. 3). Effect maps show the probability of a particular intensity (EMS-98) and the associated effects within a certain period. Magnitude maps illustrate how often an earthquake of, or above, a certain size is expected to occur within a specific radius and period.



| Hazard | Effect | Magnitude |
| --- | --- | --- |

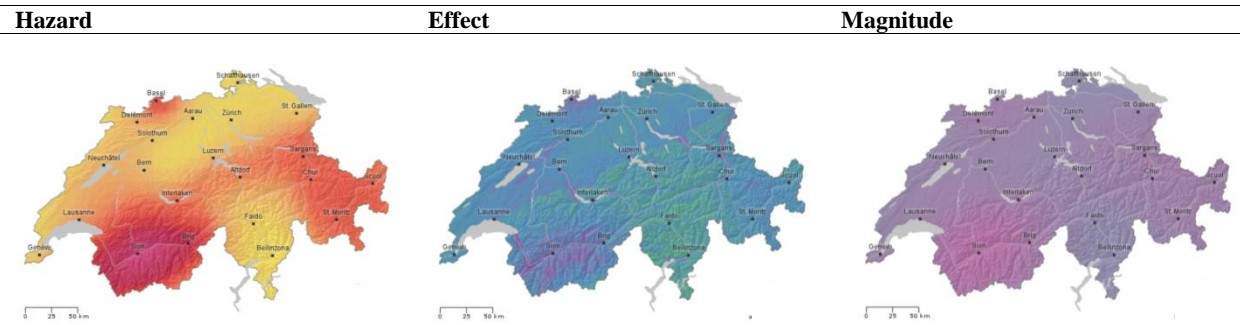

**Fig. 3.** Comparison of the color scales of the three map types offered for the release of the updated seismic hazard model: hazard map (in
units of m/sec^2), effect map (in units of EMS Intensity) and magnitude map (in units of magnitude) (Swiss Seismological Service, 2018,
www.seismo.ethz.ch/knowledge/seismic-hazard-switzerland/).

To make it easier to compare maps in terms of e.g. return periods, the same color scale is used for all map variations within
one of the three map types (Fig. 4).

| 75 years, 5 Hz | 500 years, 5 Hz | 2,500 years, 5 Hz | 10,000 years, 5 Hz |
| --- | --- | --- | --- |

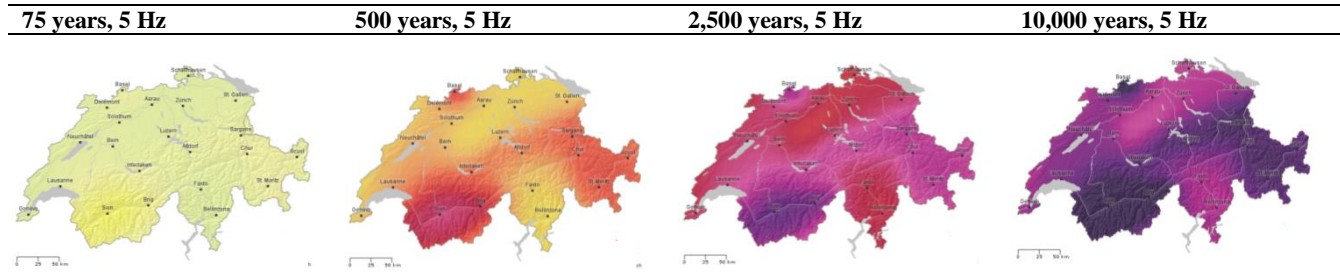

**Fig. 4.** Same color scale for hazard maps with different return periods (from left to right: 75 years, 500 years, 2,500 years, 10,000 years)
(Swiss Seismological Service, 2018, www.seismo.ethz.ch/knowledge/seismic-hazard-switzerland/maps/hazard/).



In total, 45 maps were made accessible in an interactive web tool (see Fig. 5).

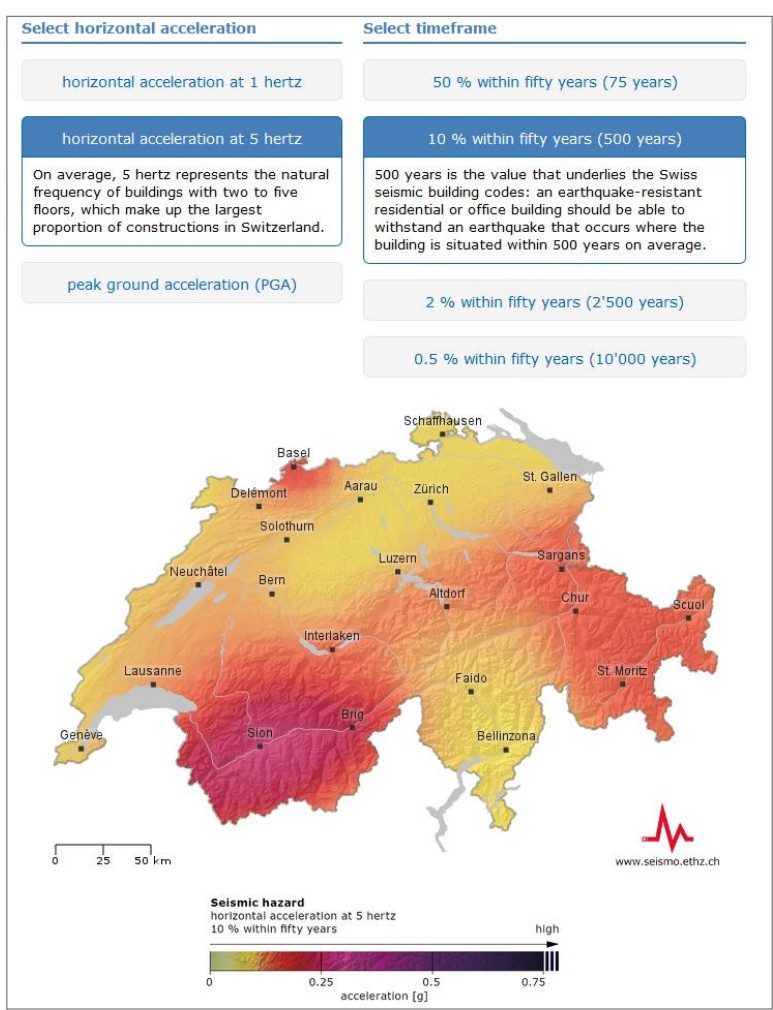

**Fig. 5.** Screenshot of the Swiss seismic hazard model interactive web tool 2015 (Swiss Seismological Service, 2018, www.seismo.ethz.ch/knowledge/seismic-hazard-switzerland/maps/hazard/).

With regard to visual characteristics (see Sect. 2.1), darker colors are used to depict areas with higher hazard, intensity, or
magnitude values, as recommended in other studies (Gaspar-Escribano and Iturrioz, 2011; Peters et al., 2008). In contrast to
its previous version (see Fig. 1, upper left corner), the seismic hazard map is mostly colored yellow to red, indicating that the
whole country is potentially endangered. All maps are unclassed as they depict continuous data, which includes the downside
of not allowing users to read single data points (Severtson and Myers, 2013). The contrast ratios are rather low, especially in
the case of the magnitude and effects map, as a consequence of using the same color scale for all maps of a certain type. Low
contrast ratios degrade the readability of the maps and also impede understanding of the information shown (Hagemeier-Klose
and Wagner, 2009; Kunz et al., 2011). The legends are prominently positioned and depict numeric and qualitative information,
as suggested by literature.



With respect to textual characteristics (see Sect. 2.2), the information provided follows best practice recommendations. Even though technical vocabulary has not been avoided in the legends, it is explained in accompanying texts around 100 to 200
words long. In addition, every map has a caption summarizing the most important parameters. All map types depict different probabilistic information, which is not only provided in numbers but also explained[2].

The interactive tool allows different map parameters to be combined individually. However, there is no option to zoom in, select specific data points or information, or personalize the map displayed (e.g. transparency), which contradicts current best practices in the field (see Sect. 2).

**3.2 Research questions**

Although there are various fragments contributing to best practices in the conceptualization of hazard maps and accompanying information, a comprehensive theoretical background is lacking. In addition, the few studies analyzing the conceptualization and comprehensibility of hazard maps mainly consulted primary users and usually worked with small, non-representative samples (Hagemeier-Klose and Wagner, 2009; Kjellgren, 2013; Meyer et al., 2012; Thompson et al., 2015). This is astonishing
considering that maps are the preferred means to communicate hazard values to a greater audience. In the absence of alternatives, they play a particularly important role in raising the awareness of the population and in influencing decisions about precautionary measures.

To fill these research gaps, our study analyzes based on a real case, how well the public, including architects and engineers not specializing in seismic retrofitting, understand and interpret the seismic hazard information provided by the SED. Our
findings will significantly depend on how well the maps are conceptualized in terms of visuals, texts, and presentation format. We are focusing on three areas: the handling and understanding of the maps, the interpretation of statistical information, and the benefit of interactive access. In addition, we are interested in factors influencing the performance of participants in understanding and interpreting hazard information, such as numeracy skills, age, gender, or education (Peters, 2008; Solberg et al., 2010; Thompson et al., 2015). Awareness and risk perception are further important precursors of future actions (Becker
et al., 2013; Lindell and Perry, 2000; Ronan and Johnston, 2005), and therefore controlled.

The most prominent output of the seismic hazard model is the seismic hazard map for a return period of 475 years. Based on the first research question, we aim to study whether people are able to correctly read and understand this particular seismic hazard map. Distinguishing hazardous from less hazardous areas requires correct interpretation of color hues, shading, and the information provided in the legend. It might also be beneficial to take into account and accurately interpret the accompanying
information.

1. Are participants able to distinguish regions with a higher seismic hazard from regions with a lower seismic hazard in Switzerland?

---

[2] For example, the term "probability of exceedance of 10 percent in 50 years (500 years)" used for the seismic hazard maps is explained in connection to the building codes: "Earthquake-resistant residential or office buildings in Switzerland are designed to withstand shaking that is expected to occur where the building is situated once every 500 years on average. The lifetime of a building is approximately fifty years. Within this lifetime, the probability of a residential or office building experiencing the design shaking is ten percent."





To execute predetermined tasks using magnitude or effect maps, participants need to derive the right conclusions based on color hues, legends, and textual information.

2.  Are participants able to choose the right magnitude or effect map for answering a given question?

3.  Are participants able to identify and correctly interpret probability values on a magnitude or effect map to answer a given question?

Statistics are fundamental for seismic hazard assessments and a genuine part of seismic hazard communication. They have proven to be very challenging to interpret. We therefore analyze how participants judge different statistical statements.

4.  How well do participants interpret statistical information?

An interactive presentation of hazard data, allowing users to answer personalized questions, is believed to support understanding of the information provided.

5.  Does interactive exploration of the Swiss seismic hazard model influence the understanding of the content provided?

## 4. Approach

A combination of quantitative and qualitative methods was chosen to best picture how people understand and interpret the maps and information offered in the context of the updated seismic hazard model for Switzerland. According to Haynes et al. (Haynes et al., 2007) quantitative methods alone fail to "capture the complexity of risk perception" in the case of volcanic hazard. When analyzing flood hazard maps, too, a combination of both approaches proved to be advantageous (Hagemeier-Klose and Wagner, 2009). We conducted an online survey of the public in order to collect data and invited architects and
engineers to participate in two workshops with a view to gaining deeper insights.

### 4.1 Sample

In total, 491 members of the public answered the online survey. Random sampling based on quotas for age, gender, education, and language was carried out by a professional research company using their panel. From a total of 1,042 participants, 478 were detained to complete the survey after having answered first sociodemographic questions. Their quota was already full.
From the remaining 564, 36 were suspended because they had not completed the survey and 37 for quality reasons because they invested less than 5 minutes to fill it in. The remaining participants took an average of 12.9 minutes to complete the survey. 257 of the participants were female and 234 male. 71.1 % filled in the German version of the online survey and 28.3 % the French version[3]. The average age of participants was 46.9 years[4] and most of them were renting a house or an apartment (66.4 %). The statistics on final examinations showed that 10.4 % had completed compulsory education, 52.1 % had gained

---

[3] Switzerland has three official languages (German, French, and Italian) and four national languages (the aforementioned three languages and Rumantsch). The survey looked at the two groups with the most representatives among the total population: German speakers (63 %) and French speakers (22.7 %) (Bundesamt für Statisik, 2015).

[4] 12.6 % of the Swiss population aged 25 to 64 have only completed compulsory education, 46.2 % hold upper-secondary-level qualifications and 41.2 % hold third-level qualifications.



upper-secondary-level qualifications (vocational education and training certificate), and 37.5 % had gained third-level
qualifications (e.g. university degree). In sum, the sample was mostly representative of the Swiss population in terms of gender,
language, and level of education.

23 architects and engineers participated in the two workshops, each of which lasted about two hours. The 4 women and 19
men were 36 years old on average and mostly worked for civil engineering companies in the German-speaking part of
Switzerland. All of them had a university degree. Participants were selected using a snowball sampling approach.

## 4.2 Procedure and measurements

Both groups, the public and the architects and engineers, started by answering a standardized questionnaire (see Table 1).
Detailed response options are specified in the Appendix. In the online survey, the response options have been randomly
reordered.

**Table 1.** First set of standardized questions. The public answered the questions online, the workshop participants on a handout. Translated
from German to English by the authors.

| First set of standardized questions | 1. Sociodemographic questions (age, gender, education etc.)<br>2. Perceived risk questions<br>3. Have you personally ever felt an earthquake in Switzerland?<br>4. How high would you classify seismic hazard in Switzerland?<br>5. Are there any areas with a particular seismic hazard in Switzerland?<br>6. Do you know the seismic hazard map the Swiss Seismological Service at ETH Zurich has published?<br>7. If so, where have you seen it?<br>8. Have you ever used this map to base on a decision? |
|---|---|

Afterwards, all participants had to conduct usability tasks (see Table 2). In the online survey, the maps questioned were always
on display, expect for question 12, where no map was depicted. All online participants had to reply the questions concerning
the hazard map for a return period of 475 years. In the following, they were randomly assigned to answer either questions
concerning the magnitude or the effect maps. For question 13 three magnitude (magnitude 5, 6, and 7) or effect (intensity IV,
VII, VIII) maps were on display. For questions 14 and 15 one map either depicting the probability for an earthquake with a
magnitude 6 or higher respectively an intensity of VIII within the next 100 years was shown.

The architects and engineers were split from the beginning into two groups. In front of a big screen, they had to navigate
through the website of the SED (www.seismo.ethz.ch) to find the information needed to solve the usability tasks. Again, all
participants were confronted with the hazard map, but only one group at a time answered the questions concerning the
magnitude respectively the effect maps. Four observers documented their discussions, their navigation paths on the website,
and their suggested answers to the given questions. Apart from the setting, the assignment of tasks was identical.




**Table 2.** Survey section with usability tasks. The public answered the questions online, the workshop participants noted their answers on a flipchart. The public solved the usability task as part of the online survey, workshop participants needed to use the SED website to find their answers. Translated from German to English by the authors.

| | | | |
|---|---|---|---|
| **Usability tasks** | Hazard map | 9. | Which are the regions with the highest seismic hazard? |
| | | 10. | Which town has the higher seismic hazard, Aarau or Interlaken? |
| | | 11. | Are there any areas in Switzerland without seismic hazard? |
| | Magnitude maps | 12a. | Which map type would you choose to answer the following question: «In which two Swiss cities is an earthquake with a magnitude of 6 or higher most likely to be expected within the next 100 years?» |
| | | 13a. | Which of the three magnitude maps depicted seems most useful to answer the following question? "In which two Swiss cities is an earthquake with a magnitude of 6 or higher most likely to be expected within the next 100 years." |
| | | 14a. | Choose the pair of cities where according to the map depicted an earthquake with a magnitude of 6 or higher has to be expected most likely. |
| | | 15a. | How big is the probability for an earthquake with a magnitude 6 or higher to occur within the next 100 years in Bern? |
| | Effect maps | 12b. | Which map type would you choose to answer the following question: "In which two Swiss cities is an earthquake causing severe damage most likely to be expected within the next 100 years?" |
| | | 13b. | Which of the three effect maps depicted seems most useful to answer the following question? "In which two Swiss cities is an earthquake causing severe damage most likely to be expected within the next 100 years." |
| | | 14b. | Choose the pair of cities where according to the map depicted an earthquake causing severe damage has to be expected most likely. |
| | | 15b. | How big is the probability for an earthquake causing severe damage to occur within the next 100 years in Bern? |

285 To conclude, both groups were asked, based on given adjectives and statements, to rate the information provided. A further set of questions dealt with their understanding of statistical information, their willingness to take precautionary measures, and amendments in their risk perception. Finally, they had to evaluate their numeracy skills (see Table 3).

**Table 3.** Second set of standardized questions. Translated from German to English by the authors.

| | | |
|---|---|---|
| **Second set of standardized questions** | 16. | What is your general impression with respect to information you have seen? |
| | 17. | How do you rate the following statements with respect to the maps you have seen before? |
| | 18. | It is mentioned several times that an event is expected "within" a certain period e.g. 50 years. What does that mean to you? |
| | 19. | Assumed, there is a 60 percent probability for a damaging earthquake at your place of living within the next 50 years. How do you rate this number? |
| | 20. | Assumed, you were living in the Valais, where there is an approximately 60 percent probability for a damaging event within 50 years. Which measures would you take to protect yourself from such an event? |
| | 21. | Have you personally taken any measures to protect yourself from the impact of an earthquake? |
| | 22. | What could we improve in the presentation of the hazard model to enhance its comprehensibility? |
| | 23. | Has your assessment of the earthquake hazard in Switzerland changed in the course of the survey? |
| | 24. | Please assess your numeracy skills by answering the following questions. |
| | 25. | Do you have additional comments about the survey? |





The following measurement parameters derive from the online survey; only there did enough people participate to allow resilient statistical statements.

A five-point Likert scale was used to measure perceived risk (question 4), covering eight items compiled in an index with a Cronbach's α of 0.753[5]. Earthquake hazard in Switzerland was classified on a five-point Likert scale (1 "very low" to 5 "very high").

In the usability section, based on a list with nine areas, we measured the number of correctly selected hazardous areas (question 9) and built a variable reflecting seismic hazard competence (see Table 4). The five areas with an elevated seismic hazard are Valais, Basel, Grisons, Central Switzerland, and Saint Gall Rhine Valley. The four areas with low to moderate seismic hazard are Jura, Tessin, Lake of Geneva Region, and Eastern Switzerland.

**Table 4.** Distribution hazard competence (N = 491).

| Hazard competence | Number of correctly selected hazardous areas | Percentages | Number |
|---|---|---|---|
| very low | 0-1 | 28.9 % | 142 |
| low | 2 | 22.8 % | 112 |
| medium | 3 | 39.9 % | 196 |
| high | 4–5 | 8.4 % | 41 |

---

[5] "Switzerland has a high earthquake hazard." / "If an earthquake hits Switzerland, major damage is to be expected." / "I do not think that a major earthquake will occur in Switzerland in the near future." / "I believe earthquakes do not pose a major threat to me." / "I am afraid that the apartment/house I am living in might be destroyed." / "I feel protected against earthquakes at my place of work." / "I feel personally affected by the earthquake hazard in Switzerland." / "Switzerland would recover fast in the aftermath of a major earthquake."



Natural Hazards
and Earth System
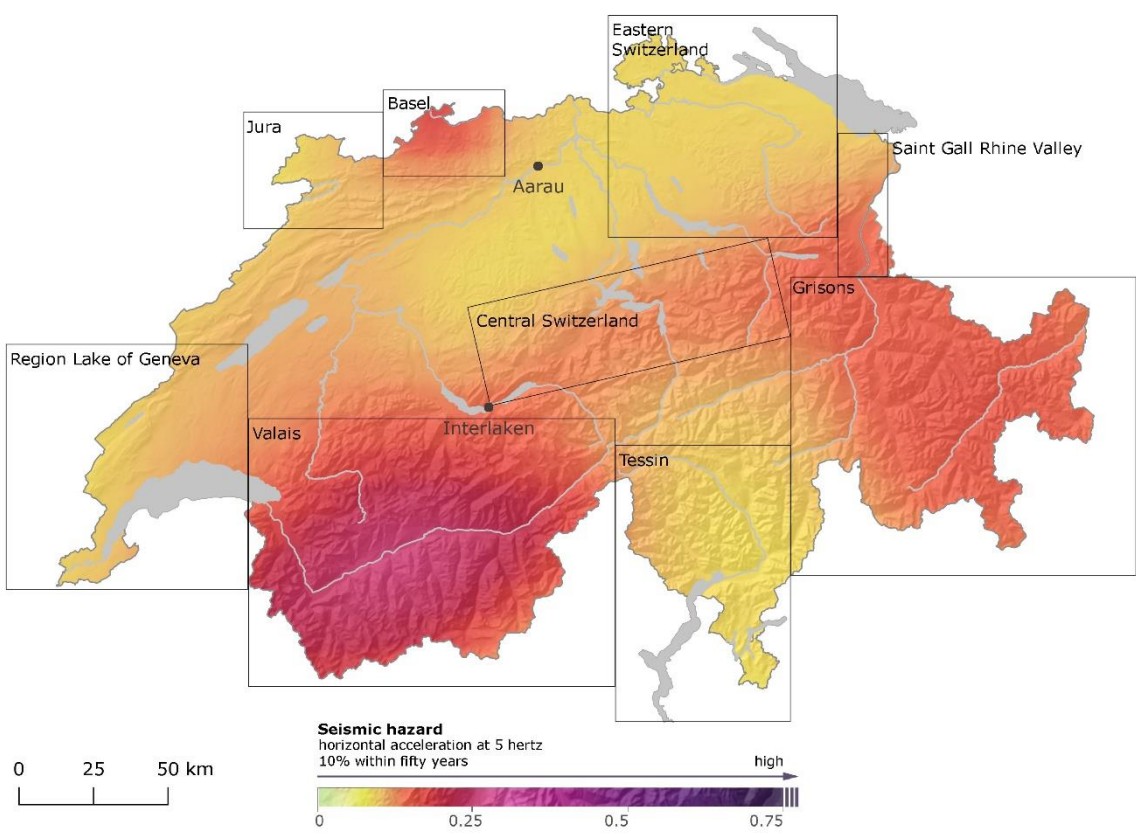

**Fig. 6.** Seismic hazard map displaying the probability of a horizontal acceleration at 5 Hertz to be experienced with 10 % within fifty years (475 years) on rocky subsoil and the areas in question (Swiss Seismological Service, 2018, www.seismo.ethz.ch/knowledge/seismic-hazard-switzerland/maps/hazard/). The map was shown to the participants without the frames highlighting the different areas.

In the second set of standardized questions we made the following measurements: Firstly, a selection of seven adjectives measuring the general impression of the information presented had to be rated on a five-point Likert scale (question 16), leading to an index with a Cronbach's α of 0.790[6]. Secondly, statements regarding the coloring of the maps, the differentiation of map types and color hues, and the explanations provided had to be rated on a five-point Likert scale; no index was compiled[7] (question 17). This was followed by two questions addressing the understanding and interpretation of statistical information, also measured on a five-point Likert scale (questions 18 and 19). To conclude, participants' numeracy skills were measured with four items[8] (Fagerlin et al., 2007) compiled in an index with a Cronbach's α of 0.916 (question 24).

---

[6] "attractive" / "trustworthy" / "helpful" / "instructive" / "complicated" / "nontransparent" / "confusing" /

[7] "The colors chosen for the maps are cumbersome to understand the information depicted." / "The difference in content the maps display is clear." / "Color differences on the various maps are not distinct enough to read out details." / "The explanations for the individual maps are comprehensive." / "The legends (captions) are helpful to understand the maps." /

[8] "How good are you at working with fractions?" / "How good are you at working with percentages?" / "How good are you at calculating a 15 % tip?" / "How good are you at figuring out how much a shirt will cost if it is 25 % off?"





## 5. Results

The following results are mainly based on the online survey conducted with members of the Swiss public. Unlike the data gathered at the workshops with the architects and engineers not specializing in seismic retrofitting, the analysis of the online survey allows for resilient statistical statements. Only the last section, concerning interactive access, solely takes into account observations made during the workshops.

### 5.1 Understanding seismic hazard maps

Before being confronted with the seismic hazard map for a return period of 475 years, most participants (85.5 %) state that they have not seen it before. With the map displayed, a majority is able to correctly select two to five hazardous areas form a total of nine regions (see Table 4). Most often, participants recognize the Valais as an area with an elevated seismic hazard, closely followed by Basel, and Grisons. As areas with an objectively lower hazard than the aforementioned regions (though they are still among the most hazardous areas in Switzerland), the Saint Gall Rhine Valley and Central Switzerland are in most cases not recognized as such (see Table 5). Almost all participants (93.5 %) successfully differentiate a city in a less hazardous area (Aarau) from one in a more hazardous area (Interlaken) (see Fig. 6). 76.4 % agree to the statement that there are not any areas without seismic hazard in Switzerland.

**Table 5.** Participants' selection of hazardous and other areas in Switzerland (N = 491) with the map displayed. Number of selections taken = 1,280.

| Areas with an elevated seismic hazard in Switzerland | Percentages of participants selecting hazardous area | Other areas | Percentages of participants selecting other area |
|---|---|---|---|
| Valais | 84.1 % | Tessin | 9.6 % |
| Basel | 60.9 % | Eastern Switzerland | 9.4 % |
| Grisons | 58.5 % | Jura | 9.4 % |
| Central Switzerland | 11.8 % | Lake of Geneva Region | 7.5 % |
| Saint Gall Rhine Valley | 9.6 % | | |
| Total selection of hazardous areas | 86.25 % | Total selections of other areas | 13.75 % |

The numeracy skills of participants significantly influence their hazard competence and how they assess the statement with respect to hazardous areas in Switzerland. Participants with advanced numeracy skills have a higher hazard competence and rather state that there are not any areas without seismic hazard in Switzerland (see Table 6).





**Table 6.** Univariate variance analysis with the numeracy skills index as dependent variable and hazard competence or areas without seismic hazard as independent variables (N = 491).

| | Index numeracy skills | | | | |
|---|---|---|---|---|---|
| **Hazard competence** | M | SD | | | |
| very low | 3.66 | 1.07 | | | |
| low | 3.75 | 1.03 | | | |
| medium | 4.02 | 0.95 | $p = 0.003$ | $F(487) = 4.70$ | $\eta^2 = 0.03$ |
| high | 4.09 | 0.91 | | | |
| **Areas without seismic hazard** | | | | | |
| Yes | 3.63 | 1.12 | $p = 0.020$ | $F(488) = 3.93$ | $\eta^2 = 0.02$ |
| No | 3.93 | 0.97 | | | |

The rating of the information provided using adjectives significantly influences participants' hazard competence as well as the choice of the city with the higher seismic hazard. Those who rate the provided information more favorable have a higher hazard competence and are more inclined to choose Interlaken instead of Aarau (see Table 7).

**Table 7.** Univariate variance analysis with the index rating of the information presented as dependent variable and hazard competence or the city pair as independent variables (N = 491).

| | Index rating of the information presented | | | | |
|---|---|---|---|---|---|
| **Hazard competence** | M | SD | | | |
| very low | 3.45 | 0.73 | | | |
| low | 3.55 | 0.71 | | | |
| medium | 3.80 | 0.67 | $p =< 0.001$ | $F(487) = 8.46$ | $\eta^2 = 0.05$ |
| high | 3.80 | 0.76 | | | |
| **City pair** | | | | | |
| Aarau | 3.13 | 0.66 | $p = 0.009$ | $F(489) = 6.80$ | $\eta^2 = 0.01$ |
| Interlaken | 3.50 | 0.79 | | | |

In response to the first research question, the majority of participants is able to correctly distinguish regions and a city with a higher seismic hazard from regions and a city with a lower seismic hazard. Furthermore, they generally assess the whole country as potentially in danger. However, only a few recognize Central Switzerland and the Saint Gall Rhine Valley as being among the areas with an elevated seismic hazard. Numeracy skills and the rating of the information provided significantly influence participants' ability to accurately identify areas or a city with an elevated seismic hazard in Switzerland.




**5.2 Understanding magnitudes and effects maps**

Participants had to select the most suitable of three magnitude or effect maps for answering a given question. The results shown in Table 8 indicate that participants make the right choice more often (see highlighted frame) when confronted with magnitude rather than effect maps.

**Table 8.** Selection of the most suitable magnitude or effect map for answering a given question. The correct answers are highlighted.

| Magnitude maps (N = 244) Map selected to answer the following question: "In which two Swiss cities is an earthquake with a magnitude of 6 or higher most likely to be expected within the next 100 years?" | | | Effect maps (N = 247) Map selected to answer the following question: "In which two Swiss cities is an earthquake causing severe damage most likely to be expected within the next 100 years?" | | |
|---|---|---|---|---|---|
| Magnitude 5 | Magnitude 6 | Magnitude 7 | Intensity VI | Intensity VII | Intensity VIII |
| 35.2 % | 56.6 % | 8.2 % | 12.2 % | 53.8 % | 34 % |


Then, with the correct magnitude or effect map displayed, participants had to select which of four city pairs has the highest probability of experiencing a magnitude 6 event or an earthquake causing severe damage. Again, they select the correct pair more often when the magnitude map was displayed ($\chi^2$ (3) = 56.72, p < 0.001) (see highlight in Table 9).

**Table 9.** Percentages of city pairs selected with maps displayed. The correct answers are highlighted.

| | Which city pair has the highest probability for experiencing a magnitude 6 event or an earthquake causing severe damage | | | |
|---|---|---|---|---|
| | Basel/Sion | Sion/Brig | Sion/Sargans | Brig/Sargans |
| **Magnitude maps** (N = 244) | 18.4 % | 71.3 % | 5.8 % | 4.5 % |
| **Effect maps** (N = 247) | 46.2 % | 39.7 % | 11.3 % | 2.8 % |


Participants perform similarly well when asked to choose the correct probability range for the occurrence of an earthquake with a magnitude of 6 or an intensity of VIII within the next 100 years in Bern ($\chi^2$ (4) = 7.73, p = 0.147) (see highlight in Table 10).

**Table 10.** Percentages of probability ranges selected with maps displayed. The correct answers are highlighted.

| | Probability range for the occurrence of an earthquake with a magnitude 6 or an intensity of VIII within the next 100 years in Bern | | | | |
|---|---|---|---|---|---|
| | 10-25 % | 25-50 % | 50-75 % | 75-100 % | Not possible to depict from the map |
| **Magnitude maps** (N = 244) | 64.8 % | 20.9 % | 6.1 % | 0.4 % | 7.8 % |
| **Effect maps** (N = 247) | 61.9 % | 18.6 % | 12.2 % | 1.2 % | 6.1 % |


As regards the second research question, the results show that participants struggle to select the most suitable of three maps for answering a given question. Their performance, especially in case of the magnitude maps, improves when asked to fulfill





a task with the right map already displayed. With respect to the third research question, results are mixed. Participants rather choose the correct city pair with the highest probability of experiencing a certain event with the magnitude map displayed. In

contrast, the probability range for a specific event in Bern is assigned equally well on the magnitude and the effect map.

### 5.3 Interpreting statistical information

Regarding the understanding of textual information describing probabilities, the statement of an event "within" a certain period of time is interpreted as intended by 73.3 % of the participants (N = 491). It can be understood as an event that has to be expected on average every 50 years, without knowing if it will happen tomorrow or in 70 years. The change of perceived risk

significantly influences the choice of the statement (see Table 11).

**Table 11.** Univariate variance analysis with the change of risk perception as dependent variable and the assessment of a verbal statement about an event within 50 years as independent variable (N = 491).

| Assessment of a verbal statement concerning an event within 50 years | Change of perceived risk | | | | |
|---|---|---|---|---|---|
| | M | SD | | | |
| certain to occur until 2067 | 3.50 | 0.99 | | | |
| certain to occur in 2067 | 3.32 | 0.67 | | | |
| to be expected on average every 50 years | 2.80 | 0.95 | $p < 0.001$ | $F(487) = 8.44$ | $\eta^2 = 0.49$ |
| I do not know | 2.85 | 0.60 | | | |

When asked to choose a verbal statement to assess the chance of a damaging earthquake occurring in their hometown within

the next 50 years with a probability of 60 percent, 72.5 % of the participants rate such an event as quite plausible or almost certain. Participants' risk perception and change of perceived risk significantly affect their assessment (see Table 12).

**Table 12.** Univariate variance analysis with the risk perception index or the change of perceived risk as dependent variable and the assessment of a verbal statement concerning an earthquake in their hometown as independent variable (N = 491).

| Assessment of a verbal statement concerning an earthquake in their hometown | Risk perception index | | | | |
|---|---|---|---|---|---|
| | M | SD | | | |
| very unlikely | 2.63 | 0.63 | | | |
| quite plausible | 2.80 | 0.67 | $p = 0.002$ | $F(488) = 6.07$ | $\eta^2 = 0.024$ |
| almost certain | 3.00 | 0.62 | | | |
| | Change of perceived risk | | | | |
| very unlikely | 2.96 | 0.76 | | | |
| quite plausible | 3.30 | 0.91 | $p < 0.001$ | $F(488) = 10.6$ | $\eta^2 = 0.04$ |
| almost certain | 3.31 | 0.70 | | | |





Risk perception and its change are important factors influencing the interpretation of statistical information. Nevertheless, over two thirds of the participants interpret the statistical statements as intended. In respect of the fourth research question, we conclude that the statistical information provided is well understood.

**5.4 Benefit of interactive access**

Only the architects and engineers not specializing in seismic retrofitting worked directly with the web tool provided by the
SED to solve the usability tasks.

The results of the observations made at the two workshops reveal that navigation through the website is challenging, at least in a group setting. The choice of the most suitable map type or map version for answering a given question proves very demanding for this group of participants too. Even though additional information was available in the form of descriptions, participants do not usually take much time to read them. They also mention that the amount of information and options to
choose from is demanding. In addition, the interpretation of single data points or probability ranges is perceived difficult. Participants criticize the web interface for not allowing them to zoom in or display specific values.

Despite having interactive access, the architects and engineers not specializing in seismic retrofitting do not differ from the participants filling in the online survey in their ability to understand and interpret the information provided. To answer the fifth research question, interactive access has no measurable positive effect.

**6. Discussion**

This study, based on a real-world setting, reveals that although communication of seismic hazard in Switzerland follows many best practice recommendations, its understanding remains challenging for the public as well as for architects and engineers not specializing in seismic retrofitting. Potential for improvements can mainly be found in the following: amount of information presented, user guidance, coloring of certain maps, and design of interactive access. As such, all elements of the map
conceptualization in terms of visuals, texts, and presentation are affected. Furthermore, personal factors like risk perception, its change, the rating of the information provided, and numeracy skills influence how hazard information is understood and interpreted.

Looking at the hazard map for a return period of 475 years, participants are generally able to differentiate areas and a city with an elevated seismic hazard from those with a lower seismic hazard. A majority also deduces correctly that there are not any
areas in Switzerland without seismic hazard. Participants' competence in handling the maps is influenced by their numeracy skills and the rating of the information provided using adjectives. The higher their numeracy skills and the better their rating of the information presented, the higher is their hazard competence. This is in line with previous findings, highlighting numeracy as an important moderator for the handling of scientific information (Keller, 2011; Peters et al., 2008; Severtson and Myers, 2013) and interpreting graphics (Spiegelhalter et al., 2011). Regarding the effect of the rating, a greater ability to read
the maps may have led to a more favorable assessment of the information presented. We conclude that the color hues chosen,





the graduation of the coloring, and the conceptualization of the legend, all of which follow best practices (see Sect. 3.1), supported the understanding of this product.

By contrast, participants are less successful in understanding and interpreting magnitude and effect maps. These additional map types were introduced to provide an alternative to the ground acceleration values depicted in hazard maps, which are usually unfamiliar to non-primary users. However, many participants struggled to select the most suitable of three maps for answering a given question. Participants would have needed to read three sentences at the bottom of each map explaining its content to make the right choice. The comparatively short average time taken to complete the online survey and the observations made at the workshops indicate that many users did not take this information into account. It is open to speculation whether three sentences already unbalance the equilibrium between completeness and comprehensibility (Peters et al., 2007) or whether the caption was just overlooked. A future study using eye tracking could shed light on this, as this method makes it possible to gain a better understanding of the elements taken into account (Keller, 2011). Furthermore, whereas for the magnitude map, the magnitude value of 6 was directly mentioned in the caption[9], the term "very severe damage" had to be autonomously translated into an intensity value of VIII [10]. Since intensity values are not commonly communicated in Switzerland, people might have struggled to understand and interpret them.

When asked to pick and interpret probability values, participants are in tendency more successful when magnitude maps, rather than effect maps, were displayed. However, a considerable amount of participants failed, which is mostly attributed to the color scales used (a criticism often brought up in the comment section and observed at the workshops). Coloring is a very sensitive component of map conceptualization (Thompson et al., 2015). As recommended for depicting continuous data, unclassified maps were compiled, which have the downside of impeding the readability of single data points (Severtson and Myers, 2013). In sum, both map types failed to apply best practices with respect to their coloring, as the shading is not sufficient (Kunz and Hurni, 2011).

The majority of participants interpret statistical information identically and as intended. Using "within" instead of "in" to describe the period for an expected event seemed to have supported the comprehensibility of the statements, as described in previous studies (Doyle et al., 2014; Hudson-Doyle et al., 2011). Two thirds further describe a damaging event occurring at their hometown with a probability of 60 as quite plausible or almost certain. Due to the semantically similarity of these options only the differentiation to the third option "very unlikely" is justifiable, which was only chosen by a minority. Participants' risk perception and its change have, at least in tendency, an effect on the interpretation of statistical information. McClure et al. (McClure et al., 2015) also show, using the example of a potential earthquake in Wellington or Christchurch, that risk perception influences likelihood estimations.

---

[9] "The map below shows the probability of an earthquake with a magnitude of 6 or higher, within a radius of 50 km, within one hundred years. In the case of earthquakes with a magnitude of 6, moderate to major damage is likely over a wide area. One hundred years represents the approximate life expectancy of a human being."

[10] "This map shows the probability of experiencing shaking on local subsoil with an intensity VIII or higher within one hundred years. In the case of an intensity VIII, major damage and even the collapse of buildings is likely. One hundred years represents the approximate life expectancy of a human being."


The two workshops conducted with architects and engineers revealed that they are similarly challenged by the tasks assigned. Interactive access had no measurable positive effect on the comprehensibility of the Swiss seismic hazard model. Besides the similar knowledge and awareness levels of engineers and architects not specializing in seismic retrofitting, the amount of information provided and the design of the interactive access may explain the outcome. As stated in other studies, too much information is rather obstructive for transmitting knowledge (Pang, 2008; Peters et al., 2007). As people only invested a little

time in going through the content, even shorter texts are advisable. With respect to interactive access, workshop participants mentioned on several occasions that the tool does not meet their expectations, which were established by use of popular commercial mapping tools (e.g. Google Maps). This attitude was also documented by Perry et al. (Perry et al., 2016). Being unable to zoom in or display specific values by clicking was seen as a major drawback and disregards best practices (Dransch et al., 2010; Hagemeier-Klose and Wagner, 2009; Kunz and Hurni, 2011). It prevents users from accessing information on

different aggregation levels, which is recommended (Kunz et al., 2011). In any case, there is always a trade-off between providing individualized information and offering too many options (Pang, 2008).

Despite some particularities of seismic hazard communication, the results of this study are transferable to any other context in which maps are used to communicate hazard to a wide range of users. The challenges observed are not limited to seismic hazard maps, but have also been observed for flood (Hagemeier-Klose and Wagner, 2009; Kjellgren, 2013) or volcanic hazard

(Thompson et al., 2015) maps and therefore apply to any other (natural) hazard. However, the results of this study are limited to Switzerland, a country with a moderate seismic hazard, and a population with low earthquake awareness.

The real-world setting brings some limitations as for example the material tested already existed. Due to the complexity of updating a hazard model, the data needed for communication materials usually only becomes available very short before the actual release. In addition, the development of communication materials is technically challenging and very resource intensive.

However, the study design mirrors adequately the setting in which seismic hazard is communicated, not only by the SED, but by many agencies around the world (see Fig. 1). Therefore, it allows for practical insights beyond theoretical considerations or lab experiments.

## 7. Conclusions and practical implications

The hazard map for a retour period of 475 years is the most prominent output of the Swiss seismic hazard model. It is frequently

requested by individuals as well as often reused by authorities and media. This map and the accompanying information follow best practices and confirm their practical use. The map seems to reach its aims by adequately informing non-experts about seismic hazard. We conclude that only when designed very carefully, natural hazard maps have the potential to also inform non-experts in the respective area.

The weak comprehensibility of magnitude and especially effect maps is attributed to the disregard of best practices. Mainly

the coloring, the impossibility of reading or accessing single data points, and the assumed unfamiliarity with intensity values impair their understanding. Improving the coloring over a wide range of values without using unappealing colors or color



combinations is very challenging. An alternative would be to classify the data (e.g. in five classes) and thus greatly simplify the map design. Further, the needs of people with visual impairments should be taken into account, an aspect not specifically evaluated and considered in the framework of this study. It is also difficult to further educate people about intensity values

without increasing the amount of information. By contrast, access to single data points could be implemented easily in an interactive tool allowing users to zoom in and click.

Finding the most appropriate information for answering questions relating to earthquake hazard has proven to be very demanding. Textual information was often not taken into account. This is a very challenging condition for the design of successful communication measures. The most obvious solution would be to improve the texts themselves, namely their

positioning and appearance, while another would be to enhance user guidance. Instead of offering all possible options at once, specific, frequently-asked questions could be answered by displaying the most suitable map automatically. As an alternative to frequently-asked questions, local scenarios (Perry et al., 2016) could be used to help people realize that such a threat is real and might impact their lives (Mileti et al., 2004; Nathe, 2000). As a result, the total of 45 maps would only be accessible in a next step for users wishing to conduct more in-depth investigations.

The deficient performance of magnitude and effect maps in particular raises the question as to whether maps are the most eligible means of communicating hazard information. Despite their extensive use there might be other, more adequate, more user-friendly means of processing the information. Infographics are currently trending as a way to communicate complex issues. They aspire to graphically represent data for a lay audience. Despite their assumed potential, there is currently only limited experimental evidence on their impact (Spiegelhalter et al., 2011). A recent analysis showed that infographics were

well received but rated as being less trustworthy (McMahon et al., 2016). Nevertheless, future studies exploring the potential of infographics to communicate seismic hazard could be beneficial.

Besides the characteristics of the information presented, users' personal traits, experiences, and perceptions influence how well they understand and interpret seismic hazard information. Risk perception and its change have proven to be of relevance, conforming previous findings: familiarity with a specific hazard is the very first step towards precautionary intentions and

actions (Whitney et al., 2004). Moreover, the effect of the change of perceived risk demonstrates that informing people is pertinent and can have an impact. However, the interplay between the information provided and personal characteristics is very complex. Since every member of a society is needed to strengthen earthquake resilience, the understanding of a regional seismic hazard is crucial for all of them. This implies that seismological services will continue to struggle to meet all users' needs when offering hazard information.

Our study shows that applying or disregarding best practices in visualization, editing, and presentation significantly impacts the comprehensibility of seismic hazard information. Due to the similarity in communicating other hazard assessments, we are convinced that our results are transferable to any other (natural) hazard context, where maps play a central role in making the results of an assessment accessible to a variety of users. We therefore strongly suggest evaluating current natural hazard communication strategies and empirically testing updated or new products. Such efforts would be of particular benefit to the

public and non-specialist professionals, who may strongly support precautionary actions.





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



## Appendix

The table below depicts the detailed response options and procedure.

**Table A1.** Detailed response options and procedure.

| | Public | Architects and engineers |
|---|---|---|
| | Online | Handout |
| **First set of standardized questions** | 1. Sociodemographic questions<br>  – Age<br>  – Gender<br>  – Leaving situation (tenant, home owner, other)<br>  – Final examinations<br>  – Canton of living<br>  – Profession<br>2. Perceived risk questions<br>  – Switzerland has a high earthquake hazard.<br>  – If an earthquake hits Switzerland, major damage is to be expected.<br>  – I do not think that a major earthquake will occur in Switzerland in the near future.<br>  – I believe earthquakes do not pose a major threat to me.<br>  – I am afraid that the apartment/house I am living in might be destroyed.<br>  – I feel protected against earthquakes at my place of work.<br>  – I feel personally affected by the earthquake hazard in Switzerland.<br>  – Switzerland would recover fast in the aftermath of a major earthquake.<br>3. Have you personally ever felt an earthquake in Switzerland?<br>  – Yes<br>  – No<br>4. How high would you classify seismic hazard in Switzerland?<br>  – 1 very low<br>  – …<br>  – 5 very high<br>5. Are there any areas with a particular seismic hazard in Switzerland?<br>  – Open section<br>6. Do you know the seismic hazard map the Swiss Seismological Service at ETH Zurich has published?<br>  – Yes<br>  – No<br>7. If so, where have you seen it?<br>  – In a printed newspaper<br>  – On the internet<br>  – In a brochure<br>  – On the website of the Swiss Seismological Service<br>  – Other<br>8. Have you ever used this map to base on a decision?<br>  – No<br>  – Yes, when buying a house<br>  – Yes, to base on a decision about insurances<br>  – Yes, as part of my work<br>  – Other | |




| | | Online | Workshop |
|---|---|---|---|
| **Usability tasks** | Hazard map | 9. Which are the regions with the highest seismic hazard?<br>– Jura<br>– Valais<br>– Grisons<br>– Central Switzerland<br>– Tessin<br>– Basel<br>– Lake of Geneva Region<br>– Eastern Switzerland<br>– Saint Gall Rhine Valley<br>10. Which town has the higher seismic hazard, Aarau or Interlaken?<br>11. Are there any areas in Switzerland without seismic hazard?<br>– Yes<br>– No | |
| | Magnitude maps | 12a. Which map type would you choose to answer the following question: «In which two Swiss cities is an earthquake with a magnitude of 6 or higher most likely to be expected within the next 100 years?"<br>– The *effects maps* focusing on potential consequences of an earthquake.<br>– The *hazard maps* depicting how often specific horizontal accelerations hit a building.<br>– The *magnitude maps* showing how often earthquakes with a specific magnitude occur.<br>13a. Which of the three magnitude maps depicted seems most useful to answer the following question? In which two Swiss cities is an earthquake with a magnitude of 6 or higher most likely to be expected within the next 100 years.<br>– The map shows the probability of an earthquake with a magnitude of 5 or higher, within a radius of 30 km, within fifty years. In the case of earthquakes with a magnitude of 6, moderate to major damage is likely over a wide area. The lifetime of the load-bearing structure of an average building is approximately fifty years.<br>– The map shows the probability of an earthquake with a magnitude of 6 or higher, within a radius of 50 km, within fifty years. In the case of earthquakes with a magnitude of 6, moderate to major damage is likely over a wide area. The lifetime of the load-bearing structure of an average building is approximately fifty years.<br>– The map shows the probability of an earthquake with a magnitude of 5 or higher, within a radius of 50 km, within fifty years. In the case of earthquakes with a magnitude of 6, moderate to major damage is likely over a wide area. The lifetime of the load-bearing structure of an average building is approximately fifty years.<br>14a. Choose the pair of cities where according to the map depicted an earthquake with a magnitude of 6 or higher has to be expected most likely.<br>– Brig & Sargans<br>– Sion & Sargans<br>– Basel & Sion<br>– Sion & Brig<br>15a. How big is the probability for an earthquake with a magnitude 6 or higher to occur within the next 100 years in Bern?<br>– 10 – 25 %<br>– 25 – 50 %<br>– 50 – 75 %<br>– 75 – 100 %<br>– Not possible to depict from the map | |


| Effect maps | 12b. Which map type would you choose to answer the following question: "In which two Swiss cities is an earthquake causing severe damage most likely to be expected within the next 100 years?"<br>– The *effects maps* focusing on potential consequences of an earthquake.<br>– The *hazard maps* depicting how often specific horizontal accelerations hit a building.<br>– The *magnitude maps* showing how often earthquakes with a specific magnitude occur.<br><br>13b. Which of the three effect maps depicted seems most useful to answer the following question? In which two Swiss cities is an earthquake causing severe damage most likely to be expected within the next 100 years.<br>– The map shows the probability of experiencing shaking on local subsoil with an intensity IV or higher within hundred years. In the case of an intensity IV, generally no damage is likely, although the earthquakes will still be felt across a wide area. One hundred years represents the approximate life expectancy of a human being.<br>– The map shows the probability of experiencing shaking on local subsoil with an intensity VII or higher within hundred years. In the case of an intensity VII, damage to buildings is likely. One hundred years represents the approximate life expectancy of a human being.<br>– The map below shows the probability of experiencing shaking on local subsoil with an intensity VIII or higher within hundred years. In the case of an intensity VIII, major damage and even the collapse of buildings is likely. One hundred years represents the approximate life expectancy of a human being.<br><br>14b. Choose the pair of cities where according to the map depicted an earthquake causing severe damage has to be expected most likely.<br>– Brig & Sargans<br>– Sion & Sargans<br>– Basel & Sion<br>– Sion & Brig<br><br>15b. How big is the probability for an earthquake causing severe damage to occur within the next 100 years in Bern?<br><br>– 10 – 25 %<br>– 25 – 50 %<br>– 50 – 75 %<br>– 75 – 100 %<br>– Not possible to depict from the map |
|---|---|
| | Online |  Handout |
| **Second set of standardized questions** | 16. What is your general impression with respect to information you have seen?<br>– attractive<br>– trustworthy<br>– helpful<br>– instructive<br>– complicated<br>– nontransparent<br>– confusing<br><br>17. How do you rate the following statements with respect to the maps you have seen before?<br>– The colors chosen for the maps are cumbersome to understand the information depicted.<br>– The difference in content the maps display is clear.<br>– Color differences on the various maps are not distinct enough to read out details.<br>– The explanations for the individual maps are comprehensive.<br>– The legends (captions) are helpful to understand the maps. |



18. It is mentioned several times that an event is expected "within" a certain period e.g. 50 years. What does that mean to you?
    – Such an earthquake will certainly occur until 2067. If not the period of 50 years has exceeded.
    – In 50 years, in 2067, such an earthquake has to be expected to occur.
    – On average, at least one such earthquake occurs over period of 50 years. It can happen tomorrow or in 70 years.
    – I do not know.

19. Assumed, there is a 60 percent probability for a damaging earthquake at your place of living within the next 50 years. How do you rate this number?
    – A damaging earthquake in the near future is very unlikely to occur at my place of living.
    – Within the next 50 years it is almost certain that a damaging earthquake occurs.
    – Within the next 50 years it is quite plausible that a damaging earthquake occurs.

20. Assumed, you were living in the Valais, where there is an approximately 60 percent probability for a damaging event within 50 years. Which measures would you take to protect yourself from such an event?
    – None
    – Earthquake-resistant construction
    – Contracting an earthquake insurance
    – Allocating an emergency food supply
    – Knowing what to do in case of an earthquake
    – Securing items inside a building e.g. shelfs
    – Other

21. Have you personally taken any measures to protect yourself from the impact of an earthquake?
    – None
    – Earthquake-resistant construction
    – Contracting an earthquake insurance
    – Allocating an emergency food supply
    – Knowing what to do in case of an earthquake
    – Securing items inside a building e.g. shelfs
    – Other

22. What could we improve in the presentation of the hazard model to enhance its comprehensibility?

23. Has your assessment of the earthquake hazard in Switzerland changed in the course of the survey?
    – 1 I assess the earthquake hazard now lower
    – …
    – 5 I asses the earthquake hazard now higher

24. Please assess your numeracy skills by answering the following questions.
    – How good are you at working with fractions?
    – How good are you at working with percentages?
    – How good are you at calculating a 15 % tip?
    – How good are you at figuring out how much a shirt will cost if it is 25 % off?

25. Do you have additional comments about the survey?
