# Peer review of "Difficulties in explaining complex issues with maps. Evaluating seismic hazard communication – the Swiss case"

_Natural Hazards and Earth System Sciences, 2019_

## Referee Comment (RC1) · Klaus Wagner (Referee) · 16 Jul 2019

In my view, this good study which evaluates seismic hazard maps has two major short-comings. 1. The researchers define which information should be extracted – it remains unclear if the maps fulfill the information needs of the target audience of the study (people at risk, architects). The authors just state: "Risk communication can lead to more accurate beliefs about seismic hazard and a higher tendency towards taking precautionary measures (Whitney et al., 2004). As elaborated previously, maps are the means of choice to communicate seismic hazard. In the following, we discuss the factors determining how hazard maps are read, interpreted, and understood. This sets

the baseline to analyze the maps 90 produced by the SED." It remains unclear which role hazard maps should play in the risk communication. In my view, there is a big difference between seismic and e.g. flood hazard maps. Flood hazard maps could easily be used by people at risk to plan mitigation measures: They get information, if their house might be flooded and the flood height for a given scenario. Thus, the information of the hazard map is mostly sufficient for the planning of mitigation measures. In contrast, seismic hazard maps cannot initiate specific mitigation measures by people at risk. The list of possible mitigation measures which is presented in question 21 can be classified in 3 categories: 1. Mitigation measures which can only be implemented by experts (Earthquake-resistant construction; Contracting an earthquake insurance) → here public only needs to know, that there is a severe danger 2. Mitigation measures which should be adopted when a basic hazard threshold is reached (Knowing what to do in case of an earthquake, Securing items inside a building e.g. shelfs) → here public only needs to know, that there is a moderate danger 3. Normal precautionary measures (Allocating an emergency food supply) → basic awareness for different types of hazards is necessary This is just a rough guess of an expert who is working on flood and alpine hazards. My expert judgement is that you are presenting to complex information which is not necessary within the overall risk communication goal (= foster private mitigation measures). Thus, my recommendation is, that you clarify the role of hazard maps within the risk communication process (you could use the path diagram of Nathe 2000 (which you have cited)). On this basis, your introduction and discussion could be improved. The second problem of the paper is in my view the data mining approach within the statistical analysis. The authors do not formulate any hypotheses, they just present statistical significant combinations they have found (ex-post) in the data set. It would be necessary that the authors formulate hypotheses on the basis of a risk perception or risk communication theory or at least a literature review. It would be interesting to present also the independent variables which had no statistical influence.

Minor comments: The authors often use following style: Meyer et al. (Meyer et al., 2012) recommend – the classical style would be: Meyer et al. (2012) l. 128: Please

remove the brackets after e.g. l. 140: ", statistical" instead of ", Statistical" l. 171: Case study instead of cases study Tables 6 and 7 are not really necessary. l. 420: Here, the studies of the Fuchs/Dorner group (2 times cited by the authors) would be helpful. This group used the eye tracking technique and could show that legends of maps were only used by expert users while lay people directly tried to interpret the map. Sorry for my limited English skills – I hope you understand my comments.

---

## Referee Comment (RC2) · Anonymous Referee #2 · 23 Jul 2019

[1] The paper "Difficulties in explaining complex issues with maps. Evaluating seismic hazard communication – the Swiss case" by Marti, Stauffacher and Wiemer, deals with the evaluation of maps as a tool to communicate seismic hazard. The maps are composed according to a set of recommendations / conditions that improve map readability and comprehensibility. The evaluation is based on the analysis and interpretation of the answers provided by different target groups to a questionnaire specifically created for this work and adapted to the Swiss case. The paper is well written, rises pertinent and interesting research questions about hazard communication and mapping features. The methodological approach to collect data followed seems adequate. Although a form with 25 questions may involve multiple analyses, the authors focus on

some specific points (INCLUDE) that lead to their key conclusions. In my opinion, this paper would be of interest for the readers of NHESS. Summing up, I recommend the publication of this paper after completing minor revisions. [2] I have a major comment on the research setup: A) The large amount of maps (45) used by the authors may hinder the comprehensibility and the ability of the respondents, as they may feel saturated of information. In my opinion, maps are good means to communicate information because they present a visual summary of information that is (or at least, should be) easy to understand. But using tens of maps makes the analysis complicated, as the reader does not distinguish the main message and may get confused by irrelevant (?) information. B) The information represented in the maps should be adapted to the end user. Specifically (and in consonance with the documentation for professionals given in the SED site): - Effects maps are risk (not hazard) maps, related to issues that any person (with any background) can observe. They are suitable for any end user. - Hazard maps are developed for rock condition (i. e. excluding site effects that could amplify ground motions) and thus give a incomplete view of the actual expected ground motions. Only specialized people (eventually including architects and engineers) would interpret these maps correctly. - Magnitude maps are basically seismicity maps, not hazard maps. I think these maps are not adequate to evaluate seismic hazard communication. I understand that the authors focus the analysis on whether the best-practice recommendations followed to elaborate the maps do facilitate hazard communication to end users. From this point of view, I have no concern with the paper. However, these points are determinant for the interpretation of results and the conclusions. Perhaps the use of a smaller amount of maps and the mapping of more user-oriented variables would lead to different conclusions. In my opinion, the issues commented in this point [2] should be included in the paper. [3] Below I provide some specific comments to the paper: 1. Introduction Lines 12-13: the usefulness of hazard maps for earthquake resistant design is mentioned as the most efficient way to reduce earthquake risk… this is valid for recent and new construction. Any comment on older (pre- seismic code) constructions? Line 17: the authors state that hazard maps "often the only accessible

information to help the public deciding about mitigation measures" and give examples in Fig. 1. There are many examples (from the countries which maps are shown in Fig. 1 among others) of other "accessible information to help people..." , maybe not maps. I would suggest this rephrasing "a principal source of information to help the public deciding about mitigation measures". 2.Best practices in communicating seismic hazard Subsection 2.2.1, line 7: I think you should add "for non-experts" at the end of "Whenever possible, technical vocabulary should be avoided" 3. Case study and focus of research Subsection 3.1, lines 25-28. Note that the hazard map is expressed in terms of acceleration and that the effects and magnitude maps are expressed in terms of probability. This may cause some confusion to the respondents. Would it be better understood an "effects (or magnitude) map" depicting the expected EMS intensity (magnitude) value for a given return period? This should be included in the discussion. Subsection 3.2, lines 20-21. The authors state "we are interested in factors influencing the performance of participants in understanding and interpreting hazard information, such as numeracy skills, age, gender or education". Please, include in the proper section an explanation about how age, gender or education influence the performance of participants in understanding and interpreting hazard information. (I SEE THAT THIS IS ALREADY TACKLED IN THE RESPONSE TO REFEREE K. WAGNER. FORGET IT). Subsection 3.2, lines 24. Please, explain what do you mean by "and therefore controlled". This sentence may require rewriting it. 4. Approach It the general public is informed about the meaning of the terms "hazard", "effects" and "magnitude" before providing the answers? How? 5. Results The first paragraph of this section is a bit confusing. Please, state how many persons of the general public and of architects/engineers constitute the sample used to assess each research question (as numbered at the end of section 3.2). If one of these research questions are answered by both groups (general public and architects/engineers), clearly indicate the differences/coincidences between the answers provided by both groups (if any). Tables: indicate the meaning of abbreviations in some tables (M for mean, SD for standard deviation, etc.) at list in one table (the first appearance). ADDITION: Sentence 425 of the

supplement to the comment of reviewer K Wagner may be confusing. Please rewrite it.

---

## Referee Comment (RC3) · Anonymous Referee #1 · 24 Jul 2019

I would like to thank the authors for their careful reply to my comments. I have to admit that I misunderstood the research question. In my view it is much more important to ask the question, if the hazard maps fulfill their role within an goal oriented risk communication (in this direction was my first major critique of the paper, although this is not the research question of the authors). Having in mind the critique of referee #2 the research question of the authors is, if the public can understand unnecessary complex information which they don't need for their preparedness actions – sorry, this formulation is a bit sarcastic, nevertheless I would like to insist a bit on my point: Especially in the conclusion section the authors could use all the empirical and expert knowledge, they have presented in the paper combined with their research results to

give recommendations which really help to improve the quality of the presented maps within a goal oriented risk communication to the public. Here the authors should consider which types of maps are used by agencies of the natural hazard management in Switzerland to inform the public about natural hazards. The most common map is the danger zone plan (Gefahrenzonenplan) which includes a risk assessment of the magnitude and frequency of different scenarios. For a good risk communication a similar risk assessment should be developed by state actor (Swiss Seismological Service, PLANAT, BAFU . . ..). Right now the Swiss Seismological Service presents many different maps which are interesting for experts but not for lay people. Here it is not helpful to give recommendations how to improve the readability of maps only experts need (e.g. lines 519ff), referee #2 talked about "irrelevant (?)" information. The information of the Swiss Seismological Service would be evaluated as "to complex" by Hagemeier-Klose and me (cited in lines 182f). Thus, the question of the conclusion section could be how the information of the service cloud be improved to initiate preparedness actions.

---

## Author Comment (AC3) · 30 Jul 2019

Dear Mr Wagner,

Thank you again for taking the time to comment our reply to your first comment.

We acknowledge your judgement of the "unnecessary complexity" of the products presented and the suggestion to simplify the design according to the danger zone plan, where the information is color coded and assigned to five categories. We agree that it would be a relevant research question to analyze if seismic hazard information integrated into such a format would be easier to understand and interpret for non-experts.

We have added a listing of potential improvements (incl. the classification of data) to the conclusion section and pointed out that these would need to be tested first to prove their usefulness in a seismic hazard context. In addition, we added the recommendation to analyze whether such amended products meet users' needs.

Most similar to such danger zone plans, but depicting data continuously, are the effect maps implemented by the Swiss Seismological Service (SED). They would best allow to deduce information about the local impact of a specific event. As specified in its report about the updated seismic hazard model, the SED introduced this map type as well as the magnitude maps because users are commonly not interested in ground acceleration values. They rather want to know how often they have to expect a damaging event or a certain magnitude in a specific area. Our results now show that these maps are less well interpreted and understood compared to the seismic hazard map. We attribute this one hand to the poor implementation of best practices and on the other on the deficient understanding of intensity. Despite the assumed value of magnitude and effect maps for a better understanding of the strength and the impact an earthquake might have, they are less requested and almost never picked-up by the media. Of course, habit could also be part of the explanation. Previously, only hazard maps were published and people might just refer to what they are more familiar with without reflecting that another product could be of more value. We have further elaborated this issue in section 3.1 and 7.

We agree that novel forms of communicating seismic hazard should be taken into consideration and also discuss potential formats e.g. infographics. We added an additional reference (Dobson et al., 2018) showing that maps lead in a flood hazard context to the least accurate decision compared with tables and graphics. In addition, we suggest to reduce the information load and to probably introduce scenarios to initiate preparedness actions. However, our study also reveals that the seismic hazard map for a return period of 475 years is well understood. Despite depicting ground acceleration values, which are unknown to most non-experts. We tried to make these aspects clearer in the

conclusion section and highlighted to increasingly evaluate users' needs.

Kind regards,

Michèle Marti (on behalf of the co-authors)

Please also note the supplement to this comment:
https://www.nat-hazards-earth-syst-sci-discuss.net/nhess-2019-112/nhess-2019-112-AC3-supplement.pdf
* * *
[Figure]

**Supplement:**

**Difficulties in explaining complex issues with maps. Evaluating seismic hazard communication – the Swiss case**

Michèle Marti[1], Michael Stauffacher[2], Stefan Wiemer[1]

[1]Swiss Seismological Service, ETH Zurich, Zurich, 8092, Switzerland
[2]USYS TdLab, ETH Zurich, Zurich, 8092, Switzerland

*Correspondence to*: Michèle Marti (michele.marti@sed.ethz.ch)

**Abstract**

2.7 billion people live in areas where earthquakes causing at least slight damage have to be expected regularly. Providing information can potentially save lives and improve the resilience of a society. Maps are an established way to illustrate natural hazard. Despite of being mainly tailored to the requirements of professional users, they are often the only accessible information to help the public deciding about mitigation measures. There is evidence that hazard maps are frequently misconceived. Visual and textual characteristics as well as the manner of presentation have been shown to influence their comprehensibility. Using a real case reflecting current practices, the material to communicate the updated seismic hazard model for Switzerland was analyzed in a representative online survey of the population (N = 491) and in two workshops involving architects and engineers not specializing in seismic retrofitting (N = 23). Although many best practice recommendations have been followed, the understanding of seismic hazard information remains challenging. Whereas most participants were able to distinguish hazardous from less hazardous areas, correctly interpreting detailed results and identifying the most suitable set of information for answering a given question proved demanding. We suggest scrutinizing current natural hazard communication strategies, and empirically testing new products, and exploring alternatives to raise awareness and enhance preparedness.

**1 Introduction**

Many of the 2.7 billion people living in areas where earthquakes causing at least slight damage have to be expected regularly[1] (Pesaresi et al., 2017) are unaware of this threat or underestimate it. Earthquake hazard is invisible as the processes of relevance occur deep underground. In addition, earthquakes are characterized as low-probability, high-impact events allowing for no warning. Currently, seismic hazard maps are the most commonly used means to visualize and communicate this danger (see a selection in Fig. 1) The preferred means of communicating complex natural hazard calculations are currently maps (Bostrom et al., 2008; Gaspar-Escribano and Iturrioz, 2011; Kunz and Hurni, 2011).
* * *
[1] The global seismic hazard map (EMMI-GSHAP) defines areas as hazardous if there is a 10 % chance of exceedance in 50 years for earthquakes with a minimal intensity of V on the Mercalli scale.

[revised manuscript text omitted]

  – Yes
  – No
7. If so, where have you seen it?
  – In a printed newspaper
  – On the internet
  – In a brochure
  – On the website of the Swiss Seismological Service
  – Other
8. Have you ever used this map to base on a decision?
  – No
  – Yes, when buying a house
  – Yes, to base on a decision about insurances
  – Yes, as part of my work
  – Other | |

| | | Online | Workshop |
|---|---|---|---|
| **Usability tasks** | Hazard map | 9. Which are the regions with the highest seismic hazard?
– Jura
– Valais
– Grisons
– Central Switzerland
– Tessin
– Basel
– Lake of Geneva Region
– Eastern Switzerland
– Saint Gall Rhine Valley

[revised manuscript text omitted]

---

## Author Response (AR2)

In my view, this good study which evaluates seismic hazard maps has two major short-comings. 1. The researchers define which information should be extracted – it remains unclear if the maps fulfill the information needs of the target audience of the study (people at risk, architects). The authors just state: "Risk communication can lead to more accurate beliefs about seismic hazard and a higher tendency towards taking precautionary measures (Whitney et al., 2004). As elaborated previously, maps are the means of choice to communicate seismic hazard. In the following, we discuss the factors determining how hazard maps are read, interpreted, and understood. This sets

the baseline to analyze the maps 90 produced by the SED." It remains unclear which role hazard maps should play in the risk communication. In my view, there is a big difference between seismic and e.g. flood hazard maps. Flood hazard maps could easily be used by people at risk to plan mitigation measures: They get information, if their house might be flooded and the flood height for a given scenario. Thus, the information of the hazard map is mostly sufficient for the planning of mitigation measures. In contrast, seismic hazard maps cannot initiate specific mitigation measures by people at risk. The list of possible mitigation measures which is presented in question 21 can be classified in 3 categories: 1. Mitigation measures which can only be implemented by experts (Earthquake-resistant construction; Contracting an earthquake insurance) → here public only needs to know, that there is a severe danger 2. Mitigation measures which should be adopted when a basic hazard threshold is reached (Knowing what to do in case of an earthquake, Securing items inside a building e.g. shelfs) → here public only needs to know, that there is a moderate danger 3. Normal precautionary measures (Allocating an emergency food supply) → basic awareness for different types of hazards is necessary This is just a rough guess of an expert who is working on flood and alpine hazards. My expert judgement is that you are presenting to complex information which is not necessary within the overall risk communication goal (= foster private mitigation measures). Thus, my recommendation is, that you clarify the role of hazard maps within the risk communication process (you could use the path diagram of Nathe 2000 (which you have cited)). On this basis, your introduction and discussion could be improved. The second problem of the paper is in my view the data mining approach within the statistical analysis. The authors do not formulate any hypotheses, they just present statistical significant combinations they have found (ex-post) in the data set. It would be necessary that the authors formulate hypotheses on the basis of a risk perception or risk communication theory or at least a literature review. It would be interesting to present also the independent variables which had no statistical influence.

Minor comments: The authors often use following style: Meyer et al. (Meyer et al., 2012) recommend – the classical style would be: Meyer et al. (2012) I. 128: Please

remove the brackets after e.g. l. 140: ", statistical" instead of ", Statistical" l. 171: Case study instead of cases study Tables 6 and 7 are not really necessary. l. 420: Here, the studies of the Fuchs/Dorner group (2 times cited by the authors) would be helpful. This group used the eye tracking technique and could show that legends of maps were only used by expert users while lay people directly tried to interpret the map. Sorry for my limited English skills – I hope you understand my comments.

**Reply to the review comment of Klaus Wagner**

Dear Mr Wagner,

Thank you very much for taking the time to review our manuscript and for your thoughtful comments.

Your first major criticism concerns the usefulness of seismic hazard maps to enhance the preparedness of a wider public. This is a very relevant objection, which is partly supported by the results of our study suggesting evaluating other means to communicate seismic hazard.

However, seismic hazard maps are currently a reality and worldwide used for this purpose. As elaborated in our paper, they are the only accessible information allowing the public to understand if they are threatened or not. They are widely requested and used by the public and decision makers. In contrast to other natural hazards, earthquake hazard is 'invisible" as the processes of relevance occur deep underground without any indication at the surface. In addition, seismic hazard is driven by low-probability but high-impact events which occur without warning. Currently, seismic hazard maps are the only established means to make this hazard visible. Thereby, seismic hazard maps play an even more important role in raising awareness compared to other natural hazard maps. Nowadays, the public as well as professionals take them into account to base on any mitigation decision. We consider it therefore as extremely important to test the use and usefulness of seismic hazard maps. We focused on the question if users are able to distinguish between hazardous and less hazardous areas and deduce further information. In this respect, they take on the same tasks as flood or other natural hazard maps.

It is true, that applying seismic design standards is the most effective mitigation measure. For this purpose, experts are needed. Nevertheless, even when strict building codes are in place, their application is often deficient or impeded. Taking Switzerland as an example, where the enforcement of building codes depends in many parts of the country exclusively on non-specialized engineers and architects or knowledgeable building owners. Currently, their only source of information allowing them to understand the seismic hazard of a given area is the information provided in the framework of the national seismic hazard map. This is also applies for home owners, who need to take a decision about contracting an earthquake insurance. This is not exclusively the case for Switzerland, but worldwide, because earthquake damages are largely uncovered. In Switzerland, this deficit is regularly debated in the national parliament and may at some point be decided by a public vote. Here also hazard maps that are understandably, transparently, and fairly portray the hazard are essential to allow the public to take an informed decision. In addition, building codes only set a minimal standard which can easily be exceed by a specific event. Therefore, individual preparedness is essential. In our opinion, earthquake preparedness does not significantly differ from other natural hazards. In any case, a knowledgeable public is needed to enforce existing regulations, to take individual measures, and to seek for professional assistance (e.g. insurance, construction work) to fill in remaining preparedness gaps.

Of course, it can and should be questioned in the future if seismic hazard maps are an adequate means to serve this purpose. Based on this real-world setting, our study is the first of its kind to analyze current approaches and thereby sets a baseline for improved hazard communication. In addition, as you correctly observed, user needs should be carefully elaborated. An aspect which

was not in the scope of our study. We therefore highlighted these aspects more clearly in the introduction and discussion sections (see supplement).

With respect to the data analysis we conducted, the parameters tested all derive from peer reviewed publications presented in Chapter 2. Based on these findings, we developed research questions to base on our analyses. In our understanding, this is the common procedure in case of poor theoretical evidence as it is the case for the evaluation of seismic hazard maps. However, based on your useful suggestion, we added and tested two hypotheses where sufficient theoretical evidence is available. In addition, we included an additional research question with respect to the currently unknown factors influencing the understanding of seismic hazard maps and also specified non-significant correlations (see supplement).

Thank you also for the minor remarks which we all considered.

Kind regards,

Michèle Marti (on behalf of the co-authors)

Nat. Hazards Earth Syst. Sci. Discuss.,
https://doi.org/10.5194/nhess-2019-112-RC2, 2019

[1] The paper "Difficulties in explaining complex issues with maps. Evaluating seismic hazard communication – the Swiss case" by Marti, Stauffacher and Wiemer, deals with the evaluation of maps as a tool to communicate seismic hazard. The maps are composed according to a set of recommendations / conditions that improve map readability and comprehensibility. The evaluation is based on the analysis and interpretation of the answers provided by different target groups to a questionnaire specifically created for this work and adapted to the Swiss case. The paper is well written, rises pertinent and interesting research questions about hazard communication and mapping features. The methodological approach to collect data followed seems adequate. Although a form with 25 questions may involve multiple analyses, the authors focus on

some specific points (INCLUDE) that lead to their key conclusions. In my opinion, this paper would be of interest for the readers of NHESS. Summing up, I recommend the publication of this paper after completing minor revisions. [2] I have a major comment on the research setup: A) The large amount of maps (45) used by the authors may hinder the comprehensibility and the ability of the respondents, as they may feel saturated of information. In my opinion, maps are good means to communicate information because they present a visual summary of information that is (or at least, should be) easy to understand. But using tens of maps makes the analysis complicated, as the reader does not distinguish the main message and may get confused by irrelevant (?) information. B) The information represented in the maps should be adapted to the end user. Specifically (and in consonance with the documentation for professionals given in the SED site): - Effects maps are risk (not hazard) maps, related to issues that any person (with any background) can observe. They are suitable for any end user. - Hazard maps are developed for rock condition (i. e. excluding site effects that could amplify ground motions) and thus give a incomplete view of the actual expected ground motions. Only specialized people (eventually including architects and engineers) would interpret these maps correctly. - Magnitude maps are basically seismicity maps, not hazard maps. I think these maps are not adequate to evaluate seismic hazard communication. I understand that the authors focus the analysis on whether the best-practice recommendations followed to elaborate the maps do facilitate hazard communication to end users. From this point of view, I have no concern with the paper. However, these points are determinant for the interpretation of results and the conclusions. Perhaps the use of a smaller amount of maps and the mapping of more user-oriented variables would lead to different conclusions. In my opinion, the issues commented in this point [2] should be included in the paper. [3] Below I provide some specific comments to the paper: 1. Introduction Lines 12-13: the usefulness of hazard maps for earthquake resistant design is mentioned as the most efficient way to reduce earthquake risk. . . this is valid for recent and new construction. Any comment on older (pre- seismic code) constructions? Line 17: the authors state that hazard maps "often the only accessible

information to help the public deciding about mitigation measures" and give examples in Fig. 1. There are many examples (from the countries which maps are shown in Fig. 1 among others) of other "accessible information to help people..." , maybe not maps. I would suggest this rephrasing "a principal source of information to help the public deciding about mitigation measures". 2.Best practices in communicating seismic hazard Subsection 2.2.1, line 7: I think you should add "for non-experts" at the end of "Whenever possible, technical vocabulary should be avoided" 3. Case study and focus of research Subsection 3.1, lines 25-28. Note that the hazard map is expressed in terms of acceleration and that the effects and magnitude maps are expressed in terms of probability. This may cause some confusion to the respondents. Would it be better understood an "effects (or magnitude) map" depicting the expected EMS intensity (magnitude) value for a given return period? This should be included in the discussion. Subsection 3.2, lines 20-21. The authors state "we are interested in factors influencing the performance of participants in understanding and interpreting hazard information, such as numeracy skills, age, gender or education". Please, include in the proper section an explanation about how age, gender or education influence the performance of participants in understanding and interpreting hazard information. (I SEE THAT THIS IS ALREADY TACKLED IN THE RESPONSE TO REFEREE K. WAGNER. FORGET IT). Subsection 3.2, lines 24. Please, explain what do you mean by "and therefore controlled". This sentence may require rewriting it. 4. Approach It the general public is informed about the meaning of the terms "hazard", "effects" and "magnitude" before providing the answers? How? 5. Results The first paragraph of this section is a bit confusing. Please, state how many persons of the general public and of architects/engineers constitute the sample used to assess each research question (as numbered at the end of section 3.2). If one of these research questions are answered by both groups (general public and architects/engineers), clearly indicate the differences/coincidences between the answers provided by both groups (if any). Tables: indicate the meaning of abbreviations in some tables (M for mean, SD for standard deviation, etc.) at list in one table (the first appearance). ADDITION: Sentence 425 of the

supplement to the comment of reviewer K Wagner may be confusing. Please rewrite it.

**Reply to the review comment of the anonymous referee #2**

Dear anonymous referee,

We very much appreciate your comprehensive acknowledgement of our paper and your helpful comments.

Your first critic focuses on the large amount of maps presented and the associated excessive demands on participants. We fully agree that especially for the workshop participants, the amount of maps accessible to solve the usability tasks was probably overwhelming as stated in the conclusion section. Nevertheless, we would like to clarify that neither the public nor the architects or engineers not specializing in seismic retrofitting were confronted with all 45 maps. In the online survey, only a selection of four different maps was presented: the seismic hazard map for a return period of 475 years and three different magnitude or effect maps. We have now specified this aspect in section 4.2. The architects and engineers had in principle access to all 45 maps. However, the seismic hazard map for a return period of 475 years is the preselected option in the interactive web tool not demanding for any further selection. In contrast, to solve the other usability tasks, different magnitudes or effect maps had to be considered. As most of the reported results are based on the online survey, we do not think that the amount of maps presented critically biases their understanding or interpretation.

Another important point you are bringing up is to reflect in more detail if the maps are meeting user requirements. In its report about the updated seismic hazard model, the SED explains that the introduction of the magnitude and effect maps was owed to the fact that users are commonly not interested in ground acceleration values. They rather want to know how often they have to expect a certain magnitude or a damaging event in a specific area. Our results now show that these maps are less well interpreted and understood compared to the seismic hazard map. We attribute this one hand to the poor implementation of best practices and on the other on the deficient understanding of intensity. Despite the assumed value of magnitude and effect maps for a better understanding of the strength and the impact an earthquake might have, they are less requested and almost never picked-up by the media. Of course, habit could also be part of the explanation. Previously, only hazard maps were published and people might just refer to what they are more familiar with without reflecting that another product could be of more value. We have further elaborated this issue in section 3.1 and 7.

With respect to your specific comments, we made the following clarifications:

(1)
Of course, respecting seismic building codes is not only important for new constructions, but also when renovating older facilities. We have added this information, thank you.

We agree that the phrasing "principle source of information" is more adequate and changed it accordingly.

(2)
We added "for non-experts" as requested.

(3)

You suggest to discuss if magnitude and effect maps would be better understood if a magnitude or intensity value would be provided for a specific return period. Currently, in the interactive web tool people can choose between three different return periods for magnitude and effect maps: one year, 50 years, and 100 years. For our study, we have chosen to only vary the magnitude and intensity values and left the return periods constant at 100 years. It would be an interesting research question to also study if different return periods affect people's understanding and interpretation of the maps.

We rewrote the sentence in section 3.2.

We specified what we mean by "therefore controlled". Because of their assumed influence all the factors previously mentioned are controlled.

(4)
The meaning of the different terms was explained in the legends included in every map depicted. We missed to mention this previously and have now added this information. In addition, a definition of every map type was provided in the selection of answers to question 12.

(5)
You are completely right; we have not explained well enough on which sample the reported results are based on. We have now specified this.

Thank you for spotting that we have not indicated the meanings of M and SD at first appearance.

We revised the misleading sentence in the last supplement.

Kind regards,

Michèle Marti (on behalf of the co-authors)

Nat. Hazards Earth Syst. Sci. Discuss.,
https://doi.org/10.5194/nhess-2019-112-RC3, 2019

[Figure]

I would like to thank the authors for their careful reply to my comments. I have to admit that I misunderstood the research question. In my view it is much more important to ask the question, if the hazard maps fulfill their role within an goal oriented risk communication (in this direction was my first major critique of the paper, although this is not the research question of the authors). Having in mind the critique of referee #2 the research question of the authors is, if the public can understand unnecessary complex information which they don't need for their preparedness actions – sorry, this formulation is a bit sarcastic, nevertheless I would like to insist a bit on my point: Especially in the conclusion section the authors could use all the empirical and expert knowledge, they have presented in the paper combined with their research results to

give recommendations which really help to improve the quality of the presented maps within a goal oriented risk communication to the public. Here the authors should consider which types of maps are used by agencies of the natural hazard management in Switzerland to inform the public about natural hazards. The most common map is the danger zone plan (Gefahrenzonenplan) which includes a risk assessment of the magnitude and frequency of different scenarios. For a good risk communication a similar risk assessment should be developed by state actor (Swiss Seismological Service, PLANAT, BAFU . . ..). Right now the Swiss Seismological Service presents many different maps which are interesting for experts but not for lay people. Here it is not helpful to give recommendations how to improve the readability of maps only experts need (e.g. lines 519ff), referee #2 talked about "irrelevant (?)" information. The information of the Swiss Seismological Service would be evaluated as "to complex" by Hagemeier-Klose and me (cited in lines 182f). Thus, the question of the conclusion section could be how the information of the service cloud be improved to initiate preparedness actions.

**Reply to the second review comment of Klaus Wagner**

Dear Mr Wagner,

Thank you again for taking the time to comment our reply to your first comment.

We acknowledge your judgement of the "unnecessary complexity" of the products presented and the suggestion to simplify the design according to the danger zone plan, where the information is color coded and assigned to five categories. We agree that it would be a relevant research question to analyze if seismic hazard information integrated into such a format would be easier to understand and interpret for non-experts. We have added a listing of potential improvements (incl. the classification of data) to the conclusion section and pointed out that these would need to be tested first to prove their usefulness in a seismic hazard context. In addition, we added the recommendation to analyze whether such amended products meet users' needs.

Most similar to such danger zone plans, but depicting data continuously, are the effect maps implemented by the Swiss Seismological Service (SED). They would best allow to deduce information about the local impact of a specific event. As specified in its report about the updated seismic hazard model, the SED introduced this map type as well as the magnitude maps because users are commonly not interested in ground acceleration values. They rather want to know how often they have to expect a damaging event or a certain magnitude in a specific area. Our results now show that these maps are less well interpreted and understood compared to the seismic hazard map. We attribute this one hand to the poor implementation of best practices and on the other on the deficient understanding of intensity. Despite the assumed value of magnitude and effect maps for a better understanding of the strength and the impact an earthquake might have, they are less requested and almost never picked-up by the media. Of course, habit could also be part of the explanation. Previously, only hazard maps were published and people might just refer to what they are more familiar with without reflecting that another product could be of more value. We have further elaborated this issue in section 3.1 and 7.

We agree that novel forms of communicating seismic hazard should be taken into consideration and also discuss potential formats e.g. infographics. We added an additional reference (Dobson *et al.*, 2018) showing that maps lead in a flood hazard context to the least accurate decision compared with tables and graphics. In addition, we suggest to reduce the information load and to probably introduce scenarios to initiate preparedness actions. However, our study also reveals that the seismic hazard map for a return period of 475 years is well understood. Despite depicting ground acceleration values, which are unknown to most non-experts. We tried to make these aspects clearer in the conclusion section and highlighted to increasingly evaluate users' needs.

Kind regards,

Michèle Marti (on behalf of the co-authors)

**Relevant changes**

Based on the very helpful comment of the anonymous reviewer and Klaus Wagner we have thoroughly revised our paper "Difficulties in explaining complex issues with maps. Evaluating seismic hazard communication – the Swiss case".

The most significant amendments concern the role and importance of non-experts in strengthening earthquake preparedness and the current relevance of seismic hazard information to base on decisions. We highlighted and discussed these aspects more prominently in the introduction, discussion, and conclusion section. In addition, we describe in more detail the potential implications of our results with respect to future approaches for communicating seismic or other natural hazards.

Another important set of changes includes more precise descriptions of the study design and meaningful additions to the data analysis.

In the following document, all changes with respect to the first version submitted are marked and therewith traceable.

In case, any further specifications about the amendments taken are required, we would be very grateful to provide additional annotations.

[revised manuscript text omitted]

   – Yes
   – No
 7. If so, where have you seen it?
   – In a printed newspaper
   – On the internet
   – In a brochure
   – On the website of the Swiss Seismological Service
   – Other
 8. Have you ever used this map to base on a decision?
   – No
   – Yes, when buying a house
   – Yes, to base on a decision about insurances
   – Yes, as part of my work
   – Other |

| | | Online | Workshop |
|---|---|---|---|
| **Usability tasks** | Hazard map | 9. Which are the regions with the highest seismic hazard?
– Jura
– Valais
– Grisons
– Central Switzerland
– Tessin
– Basel
– Lake of Geneva Region
– Eastern Switzerland
– Saint Gall Rhine Valley

[revised manuscript text omitted]

**Reply to the editor with respect to minor revisions**

The editor asked to clarify the copyrights for figures 1 and 2.

For figure 1, please find below the copyright statements of the providers of the different seismic hazard maps all allowing the reuse of the maps for non-commercial use.

a)  Swiss Seismological Service
All online documents and web pages as well as their parts are protected by copyright, and it is permissible to copy them and print them out only for private, scientific and non-commercial use.
http://www.seismo.ethz.ch/en/service/Disclaimer/

b)  USGS
USGS-authored or produced data and information are considered to be in the U.S. Public Domain. While the content of most USGS websites are in the U.S. Public Domain, not all information, illustrations, or photographs on our site are. Some non USGS photographs, images, and/or graphics that appear on USGS websites are used by the USGS with permission from the copyright holder (as required by USGS policy on use of copyrighted material). These materials are generally marked as being copyrighted. To use these copyrighted materials, you must obtain permission from the copyright holder under the copyright law.
When using information from USGS information products, publications, or websites, we ask that proper credit be given. Credit can be provided by including a citation such as the following:
Credit: U.S. Geological Survey
Department of the Interior/USGS
U.S. Geological Survey/photo by Jane Doe (if the photographer/artist is known)
Additional information on Acknowledging or Crediting USGS as Information Source is available.
If you have questions concerning the use of USGS information, please send an email to ask@usgs.gov.
https://www.usgs.gov/information-policies-and-instructions/copyrights-and-credits

c)  Bureau de recherches géologiques et minières
The copyright statement seemed ambiguous, therefore a request for reprint was sent but not answered in time. This hazard map has therefor been withdrawn from the publication.

d)  Natural Resources Canada
Non-Commercial Reproduction
• Permission to reproduce Government of Canada works, in part or in whole, and by any means, for personal or public non-commercial purposes, or for cost-recovery purposes, is not required, unless otherwise specified in the material you wish to reproduce.
• A reproduction means making a copy of information in the manner that it is originally published – the reproduction must remain as is, and must not contain any alterations whatsoever.
• The terms personal and public non-commercial purposes mean a distribution of the reproduced information either for your own purposes only, or for a distribution at large whereby no fees whatsoever will be charged.
https://www.nrcan.gc.ca/terms-and-conditions/10847

e)  INGV

Salvo diversa indicazione, il contenuto dell'intero sito è: © Osservatorio Vesuviano - INGV. La riproduzione è autorizzata solo se la fonte è citata in modo esauriente e completo.

http://www.ov.ingv.it/ov/it/copyright.html

f)  GNS

Copyright material on the Treasury website is protected by copyright owned by the Treasury on behalf of the Crown. Unless indicated otherwise for specific items or collections of content (either below or within specific items or collections), this copyright material is licensed for re-use under the Creative Commons Attribution 4.0 International licence. In essence, you are free to copy, distribute and adapt the material, as long as you attribute it to the Treasury and abide by the other licence terms. Please note that this licence does not apply to any logos, emblems and trade marks on the website or to the website's design elements. Those specific items may not be re-used without express permission.

https://treasury.govt.nz/copyright

g)  GEM

How to use and cite this work

Please cite this work as: M. Pagani, J. Garcia-Pelaez, R. Gee, K. Johnson, V. Poggi, R. Styron, G. Weatherill, M. Simionato, D. Viganò, L. Danciu, D. Monelli (2018). Global Earthquake Model (GEM) Seismic Hazard Map (version 2018.1 - December 2018), DOI: 10.13117/GEM-GLOBAL-SEISMIC-HAZARD-MAP-2018.1.This work is licensed under the terms of the Creative Commons Attribution-NonCommercial-ShareAlike 4.0 International License (CC BY-NC-SA):

https://www.globalquakemodel.org/hazard-technical-description

[Figure]

**Fig. 1.** Selection of national seismic hazard maps: a) Swiss seismic hazard map (Swiss Seismological Service, 2018, www.seismo.ethz.ch/knowledge/seismic-hazard-switzerland/) b) US seismic hazard map (United States Geological Survey, 2018, https://earthquake.usgs.gov/hazards/hazmaps/) c) Global Earthquake Model (GEM) Seismic Hazard Map (version 2018.1 - December 2018) (Pagani et al., 2018, https://www.globalquakemodel.org/gem) d) Simplified seismic hazard map for Canada, the provinces and territories (Natural Resources Canada, 2018, http://www.earthquakescanada.ca/hazard-alea/simphaz-en.php) e) Pericolosità sismica di riferimento per il territorio nazionale (Istituto Nazionale di Geofisica e Vulcanologia, 2018, http://zonesismiche.mi.ingv.it/) f) The 2010 National Seismic Hazard Model for New Zealand (Institute of Geological and Nuclear Sciences Limited, 2018, https://www.gns.cri.nz/Home/Our-Science/Natural-Hazards/Earthquakes/Earthquake-Forecast-and-Hazard-Modelling/2010-National-Seismic-Hazard-Model)

For figure 2 the legal situation is quite unclear. Screenshots are allowed as long as they do not depict an information which is "original". However, Swiss laws seem stricter and as the screenshots are all taken from Swiss media, therefor the figure has been withdrawn.